# The histone variant H2A.W and linker histone H1 co-regulate heterochromatin accessibility and DNA methylation

Pierre Bourguet [1,6], Colette L. Picard [2,6], Ramesh Yelagandula [3,5,6], Thierry Pélissier [1], Zdravko J. Lorković [3], Suhua Feng[2], Marie-Noëlle Pouch-Pélissier [1], Anna Schmücker[3], Steven E. Jacobsen [2,4], Frédéric Berger [3] & Olivier Mathieu [1✉]

In flowering plants, heterochromatin is demarcated by the histone variant H2A.W, elevated levels of the linker histone H1, and specific epigenetic modifications, such as high levels of DNA methylation at both CG and non-CG sites. How H2A.W regulates heterochromatin organization and interacts with other heterochromatic features is unclear. Here, we create a *h2a.w* null mutant via CRISPR-Cas9, *h2a.w-2*, to analyze the in vivo function of H2A.W. We find that H2A.W antagonizes deposition of H1 at heterochromatin and that non-CG methylation and accessibility are moderately decreased in *h2a.w-2* heterochromatin. Compared to H1 loss alone, combined loss of H1 and H2A.W greatly increases accessibility and facilitates non-CG DNA methylation in heterochromatin, suggesting co-regulation of heterochromatic features by H2A.W and H1. Our results suggest that H2A.W helps maintain optimal heterochromatin accessibility and DNA methylation by promoting chromatin compaction together with H1, while also inhibiting excessive H1 incorporation.

[1] CNRS, Université Clermont Auvergne, Inserm, Institut Génétique Reproduction et Développement (iGReD), Clermont-Ferrand, France. [2] Department of Molecular, Cell and Developmental Biology, University of California at Los Angeles, Los Angeles, CA, USA. [3] Gregor Mendel Institute (GMI), Austrian Academy of Sciences, Vienna BioCenter (VBC), Vienna, Austria. [4] Howard Hughes Medical Institute, University of California at Los Angeles, Los Angeles, CA, USA. [5] Present address: Institute of Molecular Biotechnology of the Austrian Academy of Sciences (IMBA), Vienna Biocenter (VBC), Vienna, Austria. [6] These authors contributed equally: Pierre Bourguet, Colette L. Picard, Ramesh Yelagandula. ✉email: olivier.mathieu@uca.fr

Eukaryotic genomes are packaged in chromatin. The basic unit of chromatin is the nucleosome, which contains a protein octamer comprising two of each of the core histones H2A, H2B, H3, and H4, wrapped by ~147 bp of DNA. Chromatin is organized into two distinct domains termed constitutive heterochromatin, which is enriched in transposable elements (TEs) and other types of repetitive DNA, and euchromatin, which comprises mostly protein-coding genes. Euchromatin is more accessible and associated with transcriptional activity, whereas heterochromatic domains prevent transcription and are often compacted into higher order structures such as chromocenters in *Arabidopsis thaliana* and mouse nuclei. Yet, heterochromatin has to retain a certain degree of accessibility to allow important DNA-related biological processes to occur, including maintenance of DNA methylation, DNA replication, DNA damage repair, and transcription for small RNA production.

In most eukaryotes, euchromatic and heterochromatic regions can be distinguished by their DNA methylation level, the presence of distinct post-translational modifications of histones, and their association with specific histone variants. In plants, DNA methylation occurs in three sequence contexts (CG, CHG, and CHH, where H is any base but G). DNA methylation in euchromatic regions tends to be low, except at CG sites over gene bodies of protein-coding genes. Heterochromatic sequences, however, are characterized by dense methylation at all three sequence contexts (CG and non-CG methylation). In *Arabidopsis*, heterochromatin is additionally decorated with histone H3 lysine 9 mono and dimethylation (H3K9me1 and H3K9me2), catalyzed by the SU(VAR)3–9 HOMOLOG-class of histone methyltransferases SUVH4/KYP, SUVH5, and SUVH6[1]. The mechanisms that establish and maintain heterochromatin-specific non-CG methylation and H3K9 methylation are tightly and reciprocally interconnected[2–4]. The DNA methyltransferases CMT2 and CMT3 are recruited by H3K9me1 and H3K9me2[2], while the H3K9 methyltransferases SUVH4/5/6 are recruited to chromatin by DNA methylation[5–7], creating a positive feedback loop that reinforces silencing. DNA methylation in plants is also established and maintained via the RNA-directed DNA methylation (RdDM) pathway, which preferentially targets short euchromatic TEs and the edges of long heterochromatic TEs[2,3,8]. Recruitment of RNA polymerase IV during the early steps of RdDM involves SHH1, which binds methylated H3K9, thereby also linking RdDM targeting to H3K9 methylation[9,10]. *Arabidopsis* heterochromatin is also marked by H3K27me1, which depends on the redundant histone methyltransferases ARABIDOPSIS TRITHORAX-RELATED 5 and 6 (ATXR5, ATXR6)[11]. Although heterochromatin structure is visibly altered in *atxr5 atxr6* mutants, DNA methylation and H3K9me2 appear largely unaffected, suggesting that H3K27me1 is maintained independently of these marks.

The linker histone H1, which binds nucleosomes and the intervening linker DNA, is also preferentially associated with heterochromatin in *Arabidopsis*[12–16]. In *Arabidopsis*, H1 associates with chromatin independently of DNA methylation, but loss of H1 leads to chromocenter decondensation, and has varying effects on DNA methylation: pericentromeric heterochromatic TEs gain DNA methylation in *h1*, while TEs on the chromosome arms lose methylation[3,14,17,18]. H1 is thought to hinder heterochromatic DNA methylation by restricting the access of DNA methyltransferases to these regions[3,19]. The joint action of H1 and CG methylation by the DNA methyltransferase MET1 silences a subset of TEs and prevents the production of aberrant gene transcripts, suggesting DNA methylation and H1 help define functional transcriptional units[14]. The histone variant H3.3 also plays a role in restricting H1 from associating with active genes in *Arabidopsis*[16], Drosophila[20], and mouse[21].

However, the mechanisms that control H1 deposition and shape its relative enrichment in heterochromatin remain unknown.

In *Arabidopsis*, the histone variant H2A.W is strictly and specifically localized to constitutive heterochromatin[22,23]. H2A variants comprise the most diverse histone family and directly impact biochemical properties of the nucleosome[24,25]. In *Arabidopsis*, nucleosomes typically contain a single type of H2A variant —either replicative H2A, H2A.X, H2A.Z, or H2A.W[26]. In land plants, the majority of H2A.Z nucleosomes associate with repressive H3K27me3 modifications while a small fraction associates with active H3K36me3 marks found at the transcription start site in transcribed genes[22,27]. Most of the body of expressed genes is occupied by replicative H2A and H2A.X nucleosomes[22]. The exclusive localization of H2A.W at constitutive heterochromatin is unique among Arabidopsis H2A variants[22,28]. While specialized histone chaperones mediate the incorporation of specific H2A variants, no chaperone dedicated to H2A.W has been identified so far[25,29]. *Arabidopsis* has three H2A.W isoforms, H2A.W.6, H2A.W.7, and H2A.W.12, encoded by *HTA6*, *HTA7*, and *HTA12* respectively. To characterize the role of H2A.W in *Arabidopsis* heterochromatin, a previous study generated a triple-knockout *hta6 hta7 hta12* line, referred here to as *h2a.w-1*[22]. The *h2a.w-1* mutants showed severely affected plant growth and were not fertile[22]. In addition, CMT3 expression and CHG methylation levels were significantly increased in *h2a.w-1*[22].

Here, we identify a large genomic rearrangement in the *hta6* transfer-DNA (T-DNA) insertion mutant allele used to generate the *h2a.w-1* triple knockout line. Using CRISPR-Cas9, we obtain a null *h2a.w* triple mutant without this rearrangement, referred to here as *h2a.w-2*. Analyzing *h2a.w-2* mutants reveals that the *hta6* chromosomal rearrangement is responsible for the severe developmental effects and CHG hypermethylation reported in *h2a.w-1*[22]. Using the mutant *h2a.w-2*, we now show that loss of H2A.W results in no visible developmental or morphological phenotypes and has only minor effects on gene and TE expression. In *h2a.w-2*, heterochromatin exhibits a mild decrease in accessibility accompanied by increased deposition of replicative H2A and H2A.X and of the linker histone H1, as well as decreased levels of non-CG methylation. The combined loss of H1 and H2A.W enhances both chromatin accessibility and DNA methylation in pericentromeric heterochromatin. Based on these results, we propose that H2A.W and H1 jointly regulate DNA methylation and heterochromatin accessibility.

## Results

**A large chromosomal translocation in *hta6* obscured functional analysis of H2A.W.** To investigate whether H2A.W plays a role in controlling TE mobilization, we used available Whole Genome Bisulfite Sequencing (BS-seq) data of *h2a.w-1* triple mutants and their corresponding wild type (WT)[22]. We detected significantly increased copy numbers for several TEs (Supplementary Table 1), but because these TEs were all located within the same genomic region on the right arm of chromosome 1, we suspected that this result reflected a chromosomal rearrangement in the *h2a.w-1* plants, rather than a genuine role for H2A.W in controlling activity of this TE subset. Further analysis of *h2a.w-1* BS-seq and RNA-seq data revealed abnormally high coverage along an approximately 5 Mb region of chromosome 1, indicating that this region may be duplicated in *h2a.w-1* (Fig. 1a, Supplementary Fig. 1a). Southern blot analysis of *hta6* SALK_024544.32 (hereafter named *hta6-1*), *hta7* (GABI_149G05.01), and *hta12* (SAIL_667_D09) DNA confirmed the presence of a genomic rearrangement, likely a translocation of part of chromosome 1, in *hta6-1* (Fig. 1b). Further analyses revealed a ~5 Mb deletion in chromosome 1 which is replaced by T-DNA/vector sequences,

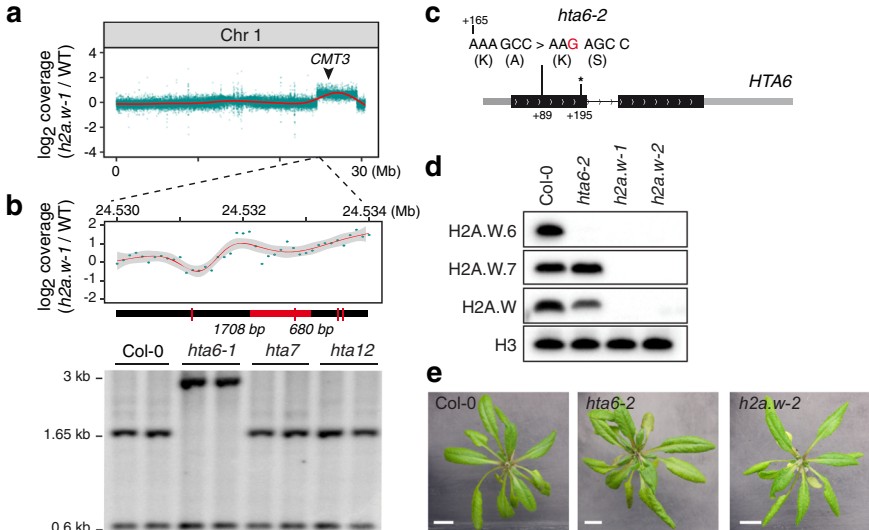

**Fig. 1 H2A.W is not required for *Arabidopsis* development. a** Sequencing coverage of published *h2a.w-1* BS-seq data[22], averaged in 1 kb bins across chromosome 1. The red line shows the smoothed conditional mean (LOESS). The black arrowhead indicates the genomic location of *CMT3*. See also Supplementary Fig. 1a. **b** Zoomed-in view of plot in **a** across the left border of the chromosome 1 region showing increased coverage in *h2a.w-1* (top panel). DNA gel blot analysis of the chromosome 1 region showing abnormal coverage in *h2a.w-1* in the indicated T-DNA insertion mutants (lower panel). Genomic DNA of the indicated genotypes was digested with *Ssp*I (recognition sites indicated by red ticks on the thick black line) and hybridized with a fragment corresponding to the genomic region indicated in red under the plot. Two independent experiments were performed with identical results. **c** The *hta6-2* CRISPR-Cas9 mutant allele. Diagram of the *HTA6* gene showing the insertion of a G (in red) in *hta6-2*, which creates a frame shift 89 bp downstream from the translation initiation site and an early stop codon (asterisk) 195 bp downstream from the translation initiation site. **d** Western blot showing total loss of H2A.W in *h2a.w-1* and *h2a.w-2*. Nuclear extracts of the indicated genotypes were analyzed using antibodies directed against H2A.W.6, H2A.W.7, total H2A.W, and H3. Two independent experiments were performed with similar results. **e** Representative images of wild-type, *hta6-2*, and *h2a.w-2* plants (scale bar = 1 cm). Both *hta6-2* and *h2a.w-2* mutants develop like wild-type Col-0 plants. Source data underlying Fig. 1b, d are provided as a Source Data file.

and a translocation of a part of chromosome 1, flanked by T-DNA sequences, to chromosome 5 (Supplementary Fig. 1b). Using segregating plants from crosses between wild-type and *hta6-1*, we were able to recover *hta6-1* mutants with either normal or doubled dosage of the chromosome 1 region. The plants with normal dosage showed a WT-like phenotype, while plants with doubled dosage were abnormally small (Supplementary Fig. 1b), suggesting that increased dosage of this portion of chromosome 1 causes developmental defects.

Using targeted mutagenesis via CRISPR-Cas9, we generated a *hta6* allele (*hta6-2*) carrying a single-base frame shift mutation that causes a stop codon early in the protein (Fig. 1c). Western blot confirmed that *hta6-2* is a null mutant for H2A.W.6 (Fig. 1d). We crossed *hta6-2* with *hta7* and *hta12* to obtain the null triple mutant, *h2a.w-2*. Western blot analysis confirmed that H2A.W is completely absent in *h2a.w-2* (Fig. 1d and Supplementary Fig. 1c). *h2a.w-2* plants are morphologically indistinguishable from wild-type plants (Fig. 1e), indicating that increased dosage of a large portion of chromosome 1, and not loss of H2A.W, caused the strong developmental defects reported in *h2a.w-1*[22]. Instead, H2A.W appeared to be dispensable for *Arabidopsis* development. We therefore sought to clarify the function of H2A.W in heterochromatin transcription, composition, and organization.

**H2A.W has little impact on transcription but is required for the efficient methylation of heterochromatic DNA.** Previous analyses of the impact of H2A.W loss on transcription may have been confounded by the genomic rearrangement in *h2a.w-1*. We therefore re-explored whether lack of H2A.W affects genome-wide transcription by performing RNA-seq of WT and *h2a.w-2*. These analyses identified only a few differentially expressed protein-coding genes (PCGs; 78 upregulated, 52 downregulated) and only a handful of transcriptionally activated TEs in *h2a.w-2*

(Supplementary Fig. 2a, b). These results show that gene expression is not strongly affected in the absence of H2A.W, and that H2A.W either does not play a significant role in repressing TEs or that other pathways are compensating for H2A.W loss.

The chromosomal rearrangement in *h2a.w-1* caused a duplication of the gene encoding CMT3 (Fig. 1a, Supplementary Fig. 1a) and this duplication was likely responsible for the higher levels of CHG methylation previously reported in *h2a.w-1*[22]. Indeed, we found that *CMT3* mRNA was significantly increased in *hta6-1* and in *h2a.w-1* carrying additional *CMT3* gene copies (Supplementary Fig. 2c). However, *CMT3* expression in *h2a.w-2* was similar to WT (Supplementary Fig. 2c), indicating that *CMT3* transcription is not affected by loss of H2A.W. We were also able to recover *h2a.w-2 cmt3* quadruple mutant plants, which had no obvious developmental defects (Supplementary Fig. 2d), indicating that the lethal genetic interaction between *h2a.w-1* and *cmt3*[22] was due to the chromosomal rearrangement in *hta6-1*.

We therefore sought to clarify the impact of H2A.W loss on DNA methylation by examining the methylome of *h2a.w-2* using BS-seq. We found no conspicuous change in CG DNA methylation in the *h2a.w-2* mutant (Supplementary Fig. 3a, b), while non-CG methylation levels appeared substantially decreased at pericentromeric regions (Supplementary Fig. 3a). In agreement with these chromosome-wide observations, TEs located in the pericentromeres showed partially reduced CHG and CHH methylation levels (Fig. 2a, Supplementary Fig. 3c), suggesting that H2A.W is required for maintenance of DNA methylation in these regions. Conversely, we found that TEs located on chromosome arms showed increased CHH DNA methylation in *h2a.w-2*, suggesting an antagonistic effect of H2A.W (Fig. 2a, Supplementary Fig. 3c). Indeed, looking specifically at regions normally occupied by H2A.W in WT revealed opposing changes in non-CG DNA methylation in the *h2a.w-2* mutant, based on chromosomal location. A substantial decrease in non-CG methylation levels was observed over H2A.W

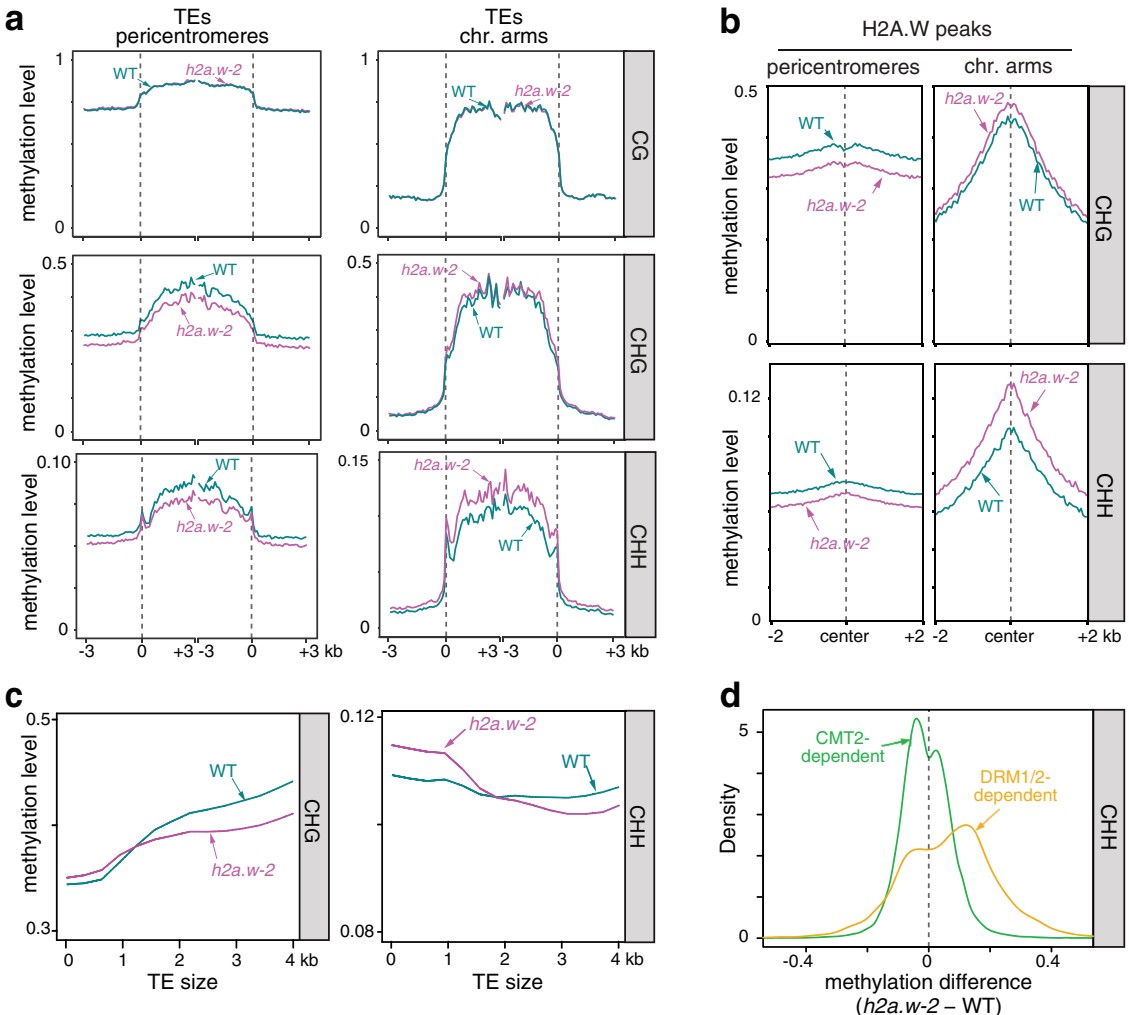

**Fig. 2 Loss of H2A.W alters non-CG DNA methylation patterns. a** CG, CHG, and CHH methylation levels over TEs in pericentromeres and chromosome arms in WT and *h2a.w-2*. TEs were aligned at the 5′ (left dashed line) and 3′ end (right dashed line), and sequences 3 kb upstream or downstream were included, respectively. Average methylation over 100 bp bins is plotted. **b** CHG and CHH methylation levels over H2A.W peaks in the chromosome arms and pericentromeres. **c** Locally weighted scatterplot smoothing (LOESS) fit of CHG and CHH methylation levels in WT and *h2a.w-2* calculated in 50 bp windows and plotted against TE size. **d** Kernel density plots of CHH DNA methylation differences between *h2a.w-2* and WT at CMT2-dependent and DRM1/2-dependent regions.

peaks in pericentromeric regions, whereas regions located in chromosome arms showed increased methylation levels (Fig. 2b, Supplementary Fig. 3d). Short TEs enriched in chromosome arms are known targets of the RdDM pathway involving DRM1/2, while CHG and CHH methylation at long heterochromatic TEs is preferentially maintained by CMT3 and CMT2, respectively[2,3]. We found that short TEs tended to gain CHH methylation in *h2a.w-2*, while long heterochromatic TEs instead tended to lose CHH and CHG methylation (Fig. 2c). Accordingly, non-CG methylation was reduced at CMT2-dependent regions but increased at DRM1/2-dependent regions in *h2a.w-2* (Fig. 2d). Together, these findings indicate that H2A.W promotes CMT3 and/or CMT2-mediated methylation maintenance in pericentromeric heterochromatin. However, H2A.W incorporation does not require CHG methylation[22]. Our data also suggest that H2A.W opposes RdDM at less heterochromatic regions on chromosome arms, supporting the idea that RdDM is inhibited by heterochromatin as proposed previously[3,30].

**Heterochromatin accessibility decreases in the absence of H2A. W.** Since complex and global changes in patterns of DNA

methylation are often associated with modulation of chromatin accessibility[19], we analyzed chromatin organization in *h2a.w-2*. Indeed, we observed enlarged chromocenters in *h2a.w-2* (Fig. 3a), as was previously observed in *h2a.w-1*[22]. Chromocenter enlargement is also observed in mutants that induce over-replication of heterochromatic regions[31]. To test for over-replication, we analyzed DNA content in *h2a.w-2* nuclei by FACS. We did not observe any obvious change relative to WT (Supplementary Fig. 4), indicating that enlargement of chromocenters most likely reflected changes in chromatin organization caused by loss of H2A.W, as reported previously[22]. By introducing a H2A.W.6 genomic construct in *h2a.w-2* mutants, we could partially rescue the enlargement of chromocenters in *h2a.w-2* mutants, confirming that H2A.W regulates chromocenter formation (Supplementary Fig. 5a, b). Similarly, protein coding genes upregulated in *h2a.w-2* had diminished transcript levels in mutants complemented with a H2A.W.6 genomic construct (Supplementary Fig. 5c).

To investigate the impact of H2A.W on chromatin accessibility, we applied the Assay for Transposase Accessible Chromatin using sequencing (ATAC-seq)[32] to *h2a.w-2* and WT ten-day-

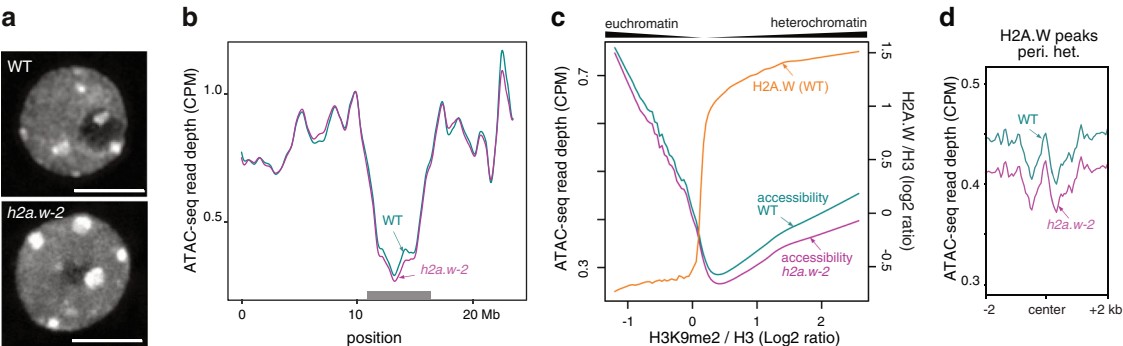

**Fig. 3 Loss of H2A.W alters heterochromatin organization and accessibility. a** Representative images of DAPI-stained WT (Col-0) and *h2a.w-2* leaf interphase nuclei. Two independent experiments were performed with similar results. Scale bar is 5 μm. **b** Locally weighted scatterplot smoothing (LOESS) fit of ATAC-seq read depth averaged in 1 kb bins across chromosome 3 in WT and *h2a.w-2*. Average of two replicates shown. The gray rectangle indicates the location of pericentromeric heterochromatin. **c** Smoothing spline fit (50 degrees of freedom) of WT H2A.W levels (log$_2$ ChIP-seq H2A.W/H3) and of ATAC-seq read depth (CPM normalized) in WT and *h2a.w-2* in 1 kb windows plotted against WT H3K9me2 level. **d** ATAC-seq read depth over H2A.W peaks in pericentromeric heterochromatin. Average of two replicates shown.

old seedlings. In WT, the accessibility of pericentromeric chromatin and regions associated with H2A.W was low relative to other regions, as expected (Fig. 3b–d, Supplementary Fig. 6a, b). Interestingly, in heterochromatic regions (those with high levels of H2A.W and H3K9me2), chromatin accessibility was positively correlated with levels of H2A.W and H3K9me2, so that highly heterochromatic sequences were relatively more accessible than less heterochromatic sequences (Fig. 3c). Unexpectedly, we observed a modest reduction in heterochromatin accessibility in *h2a.w-2* that correlated well with WT H2A.W levels, suggesting H2A.W provides heterochromatin with a certain degree of accessibility in the WT (Fig. 3b–d and Supplementary Fig. 6a, b). This observation was at odds with previous reports showing that H2A.W promotes nucleosome thermostability[26] and compaction of nucleosome arrays[22], suggesting that the decreased accessibility observed in *h2a.w-2* heterochromatin may not be the direct consequence of the loss of H2A.W. We hypothesized that the reduction of chromatin accessibility in *h2a.w-2* results from the replacement of H2A.W by other types of H2A variants and/or the deposition of a chromatin component that impedes chromatin accessibility.

**H2A.X and replicative H2A replace H2A.W in *h2a.w-2* heterochromatin**. To assess the composition of chromatin in *h2a.w-2*, we profiled the genome-wide distribution of H3 and other H2A variants by chromatin immunoprecipitation followed by high-throughput sequencing (ChIP-seq). Profiles of H3 enrichment determined by ChIP-seq were similar in WT and *h2a.w-2* plants, suggesting that nucleosome density was not responsible for the change in accessibility in *h2a.w-2* (Supplementary Fig. 7a). Since H2A variants confer distinct stability to nucleosomes[26], the replacement of H2A.W by another H2A variant is expected to affect chromatin properties. Based on in vitro thermostability assays, replicative H2A confers higher stability than H2A.W and H2A.X, whereas H2A.Z nucleosomes are the least stable[26]. Similarly, different H2A variants are more or less favorable to the deposition of DNA methylation[28]. Therefore, we explored the distribution of the other three H2A variants in *h2a.w-2* plants by profiling H2A.X (H2A.X.3 and H2A.X.5), H2A.Z (H2A.Z.9), and replicative H2A (H2A.1 and H2A.13) using ChIP-seq. In WT, H2A.Z, H2A.X, and H2A showed relative depletion over pericentromeric heterochromatin and at H2A.W-associated regions (Fig. 4a, b), as previously reported[22]. In *h2a.w-2*, we found a striking gain of H2A.X, and to a lesser extent replicative H2A, but not H2A.Z, over regions normally marked by H2A.W in WT

(Fig. 4a–c). CG DNA methylation has been shown to exclude H2A.Z from methylated DNA, and H2A.Z and CG methylation are largely anticorrelated in WT plants[28,33]. The largely unchanged H2A.Z distribution in *h2a.w-2* is consistent with our observation that CG methylation patterns are also unaltered in *h2a.w-2* (Fig. 2a, Supplementary Fig 3a, b). Western blot analyses confirmed increased levels of replicative H2A and H2A.X in *h2a.w-2* chromatin (Supplementary Fig. 7b).

Although replicative H2A nucleosomes show higher thermal stability than H2A.W nucleosomes, in vitro DNA protection assays have shown that replicative H2A confers less protection than H2A.W[26]. This indicated to us that the increase in replicative H2A at pericentromeric chromatin may not be responsible for the observed decrease in accessibility. The in vitro thermostability of H2A.W and H2A.X nucleosomes is similar[26] and thus also should not directly account for the changes in chromatin accessibility in *h2a.w-2*. However, H2A.X is primarily known for its role in DNA damage response, during which it becomes rapidly phosphorylated to form γH2A.X aggregates[23], which could impact chromatin accessibility. The ratio of γH2A.X/H2A.X remained unchanged in *h2a.w-2* (Supplementary Fig. 7b), suggesting that γH2A.X was not responsible for the change in chromatin accessibility in *h2a.w-2*. In support of this conclusion, DNA damage response genes are not mis-regulated in *h2a.w-2* (Supplementary Fig. 7c).

As changes in H2A variant composition were unlikely to be directly responsible for the decreased heterochromatin accessibility in *h2a.w-2*, we next examined other epigenetic modifications. Profiles of epigenetic marks typically associated with heterochromatin, namely H3K9me1, H3K9me2, and H3K27me1, were similar in WT and *h2a.w-2* (Supplementary Fig. 7a). Hence, maintenance of these post translational modifications is independent of H2A.W and these modifications are not responsible for the change in heterochromatin accessibility in *h2a.w-2*. Intriguingly, reduced levels of non-CG methylation in *h2aw-2* were not accompanied by detectable changes in H3K9me2, suggesting that maintenance of H3K9me2 by SUVH4/5/6 may be less sensitive than CMT2 and CMT3 activities to changes in accessibility occurring in *h2a.w-2* (Supplementary Fig. 7a). Alternatively, changes in H3K9me2 could be below the detection threshold allowed by ChIP-seq.

**Heterochromatin H1 levels increase in *h2a.w-2***. Interestingly, the DNA methylation changes in *h2a.w-2* appeared to be the inverse of those previously found in linker histone H1 mutants, which display decreased DNA methylation at short euchromatic

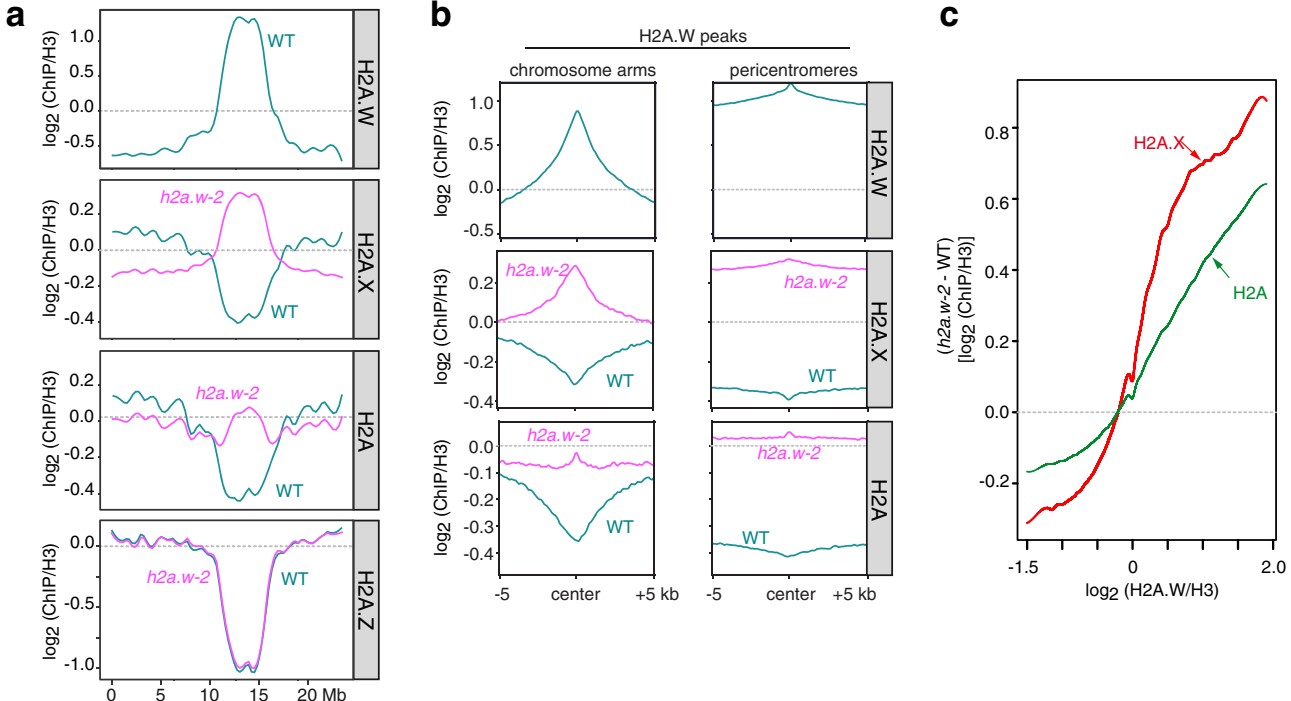

**Fig. 4 Replicative H2A and H2A.X replace H2A.W in *h2a.w-2*. a** Locally weighted scatterplot smoothing (LOESS) fit of H2A.W, H2A.X, replicative H2A and H2A.Z levels averaged in 1 kb bins across chromosome 3 in WT and *h2a.w-2*. Average of two replicates shown. **b** Metaplots of average H2A.W, H2A.X, and replicative H2A levels from two replicates over H2A.W peaks in the chromosome arms and pericentromeres. **c** Smoothing spline fits (50 degrees of freedom) of changes in H2A.X and replicative H2A levels (*h2a.w-2* minus WT; log₂ ChIP/H3) in 1 kb windows plotted against WT H2A.W level. Source data underlying Fig. 4a are provided as a Source Data file.

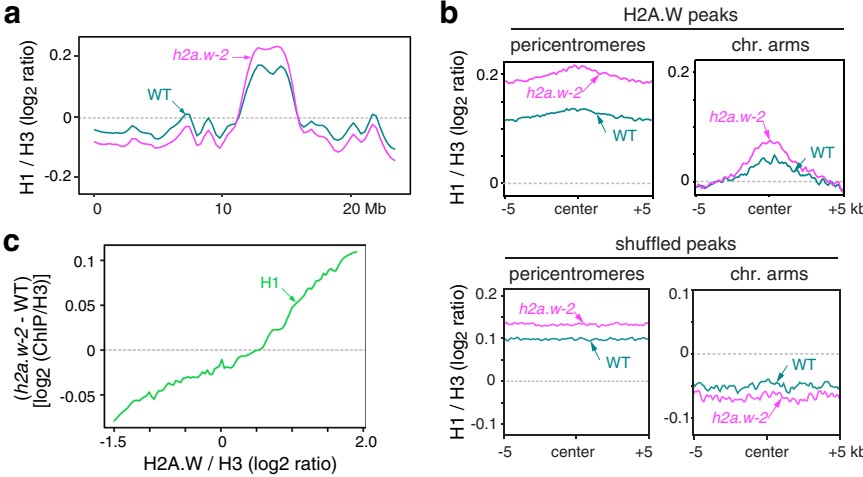

**Fig. 5 H2A.W antagonizes histone H1 deposition in heterochromatin. a** Locally weighted scatterplot smoothing (LOESS) fit of H1 levels averaged in 1 kb bins across chromosome 3 in WT and *h2a.w-2*. Average of two replicates shown. **b** Metaplots of average H1 levels from two replicates over H2A.W peaks in pericentromeres and the chromosome arms (top panel). Plots over randomly shuffled peaks within the chromosome arms and pericentromeres are shown for comparison (bottom panel). **c** Smoothing spline fits (50 degrees of freedom) of change in H1 levels (*h2a.w-2* minus WT; log₂ ChIP/H3) in 1 kb windows plotted against WT H2A.W level. Source data underlying Fig. 5a are provided as a Source Data file.

TEs and increased methylation at long heterochromatic TEs[3]. This suggested that the DNA methylation changes in *h2a.w-2* may be related to changes in H1 distribution and prompted us to explore H1 patterns in *h2a.w-2*. Consistent with earlier work[14–16], our ChIP-seq analyses revealed that H1 is enriched in pericentromeric heterochromatin relative to euchromatin in WT (Fig. 5a). Regions associated with H2A.W in WT were also generally enriched in H1 (Fig. 5b). In *h2a.w-2*, we observed a further increase in H1 at pericentromeric heterochromatin,

accompanied by a modest decrease of H1 along chromosome arms (Fig. 5a). Regions normally marked by H2A.W in both pericentromeric regions and chromosome arms also showed increased H1 enrichment in *h2a.w-2* relative to WT that correlated well with WT H2A.W levels (Fig. 5b, c), indicating that H2A.W opposes deposition of H1. Western blot analysis indicated that global nuclear H1 levels were similar in *h2a.w-2* and WT (Supplementary Fig. 7b), suggesting that the total pool of H1 available is limiting and that increased recruitment of H1 in

pericentromeric heterochromatin and other H2A.W-associated regions in *h2a.w-2* is likely responsible for the relative depletion of H1 along the chromosome arms (Fig. 5a, b). Histone H1 is known to obstruct chromatin accessibility and stabilize nucleosomes by binding to the linker DNA[12]. Therefore, redistribution of H1 may account for the change in chromatin accessibility in *h2a.w-2* (Fig. 3b–d, Supplementary Fig. 6a, b).

**H2A.W and H1 cooperatively control heterochromatin accessibility.** To test whether increased H1 levels could explain the reduced accessibility and DNA methylation in *h2a.w-2* pericentromeres, we obtained *h2a.w-2 h1.1 h1.2* quintuple mutants (hereafter named *h1 h2a.w*), which lack both H2A.W and H1 (Supplementary Fig. 6c). We then compared chromatin accessibility and DNA methylation between *h2a.w-2*, *h1* and *h1 h2a.w* mutants. Chromocenters were dispersed in *h1* mutant nuclei, consistent with recent reports[14,18], and even further dispersed in *h1 h2a.w* (Fig. 6a, b). Consistent with this, chromatin accessibility increased over pericentromeric heterochromatin regions

associated with H2A.W in *h1* and further in *h1 h2a.w* (Fig. 6c, d, Supplementary Fig. 6a, b). This is in agreement with recent MNase-seq profiles showing reduced nucleosomal density in *h1* mutants[18]. Moving from chromosomal arms into pericentromeric regions, heterochromatin content and average TE length increase,[3] and H2A.W levels are well-correlated with TE length in WT (Fig. 6e). Gain in chromatin accessibility in *h1* was also correlated with TE length and WT H2A.W levels, essentially mirroring changes in accessibility in *h2a.w-2* (Fig. 6e). Combined loss of H1 and H2A.W in *h1 h2aw* mutants led to an even stronger increase in chromatin accessibility at regions normally associated with H2A.W (Fig. 6e). This indicates that H2A.W restricts heterochromatin accessibility in the absence of H1.

DNA methylation profiles of *h1* mutants were consistent with previously published data, showing increased methylation over long heterochromatic TEs and reduced CHH methylation at short TEs (Fig. 7a and Supplementary Fig. 8)[3]. Similarly, non-CG methylation levels increased at CMT2-dependent regions but were reduced at DRM2-dependent regions in *h1* (Fig. 7b, c). Again, these changes were largely opposite to those in *h2a.w-2*

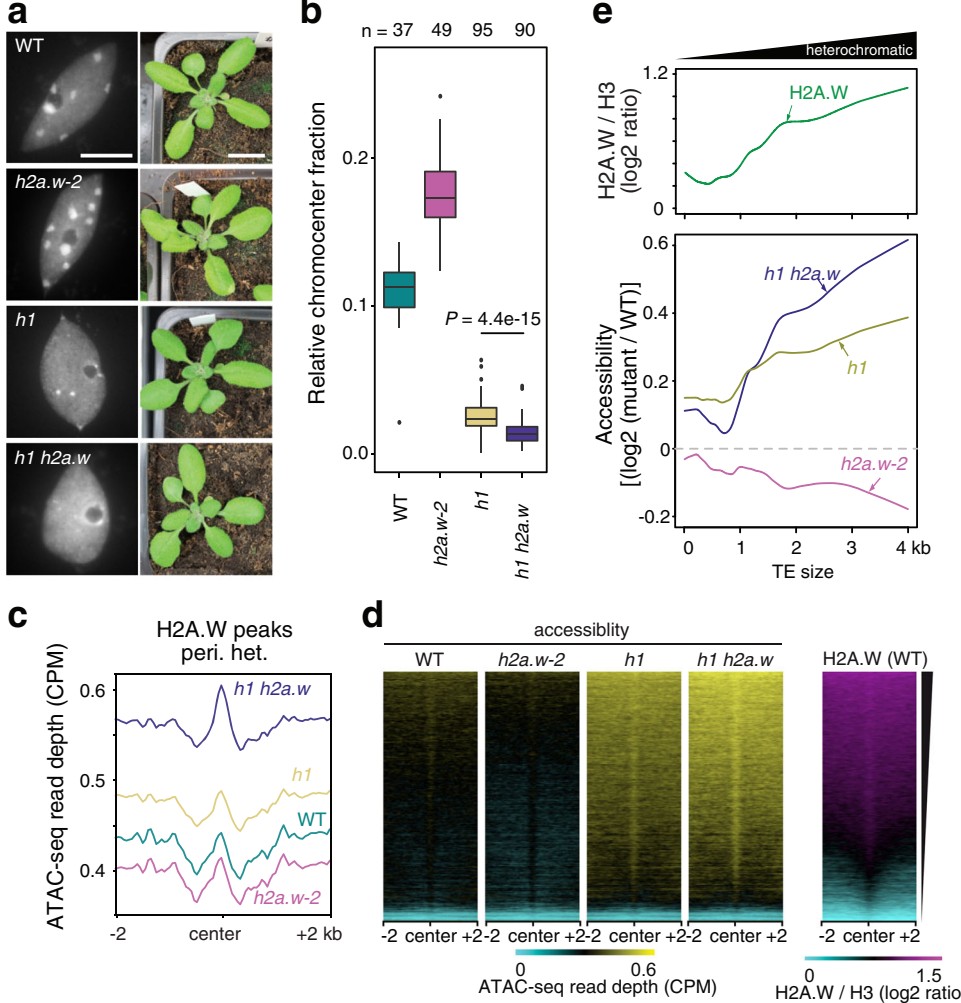

**Fig. 6 H2A.W and H1 co-regulate heterochromatin accessibility. a** Representative images of DAPI-stained WT (Col-0), *h2a.w-2*, *h1* and *h1 h2a.w* leaf interphase nuclei (left; scale bar is 5 μm) and of 3-week-old plants of the same genotypes (right; scale bar is 1 cm). Nuclear preparation and analysis were independently repeated three times with similar results. **b** Quantification of the relative chromocenter fraction in WT (Col-0), *h2a.w-2*, *h1* and *h1 h2a.w* nuclei. Number of analyzed nuclei are indicated on the top. Whiskers indicate 1.5X IQR. Outliers are represented by circles. Relative chromocenter fraction in *h1* and *h1h2a.w* show statistically significant difference (*P* = 4.4e-15, unpaired Mann–Whitney test). **c** ATAC-seq read depth over H2A.W peaks in pericentromeric heterochromatin. Average of two replicates shown. **d** Heat maps of average ATAC-seq read depth over H2A.W peaks in pericentromeres ranked based on level H2A.W signal in WT. **e** Locally weighted scatterplot smoothing (LOESS) fit of WT H2A.W levels (log₂ ChIP/H3; top panel) and changes in chromatin accessibility (mutant / WT; ATAC-seq read depth) in 50 bp windows plotted against TE size.

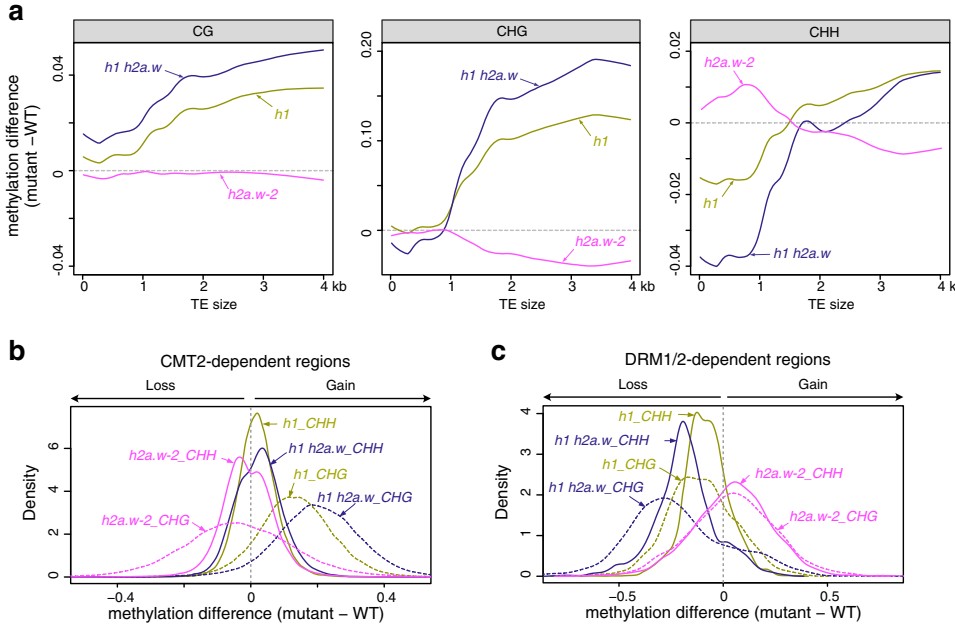

**Fig. 7 Impact of combined loss of H2A.W and H1 on DNA methylation. a** Locally weighted scatterplot smoothing (LOESS) fit of changes in CG, CHG, and CHH DNA methylation in *h2a.w-2*, *h1* and *h1 h2a.w* mutants (mutant minus WT) in 50 bp windows plotted against TE size. **b** Kernel density plots of CHG and CHH DNA methylation changes at CMT2-dependent regions in *h2a.w-2*, *h1* and *h1 h2a.w* mutants. **c** Kernel density plots of CHG and CHH DNA methylation changes at DRM1/2-dependent regions in *h2a.w-2*, *h1* and *h1 h2a.w* mutants.

mutants, supporting the hypothesis that H1 is responsible for altered DNA methylation in *h2a.w-2*. H1 has been shown to control DNA methylation over heterochromatin, presumably by restricting access of DNA methyltransferases to heterochromatic DNA[3]. In concordance with increased heterochromatin accessibility in *h1 h2a.w*, we found that long heterochromatic TEs and CMT2-dependent regions further gained DNA methylation in *h1 h2a.w* compared to *h1* alone (Fig. 7b). This supports a positive correlation between heterochromatin accessibility and DNA methylation levels, and suggests that H2A.W hinders access of DNA methyltransferases to heterochromatic DNA in the absence of H1.

Overall, pericentromeres gain H1 and lose both accessibility and non-CG methylation in *h2a.w-2* and these changes are strongly reversed in *h1 h2a.w* mutants (Fig. 8). This is consistent with a model wherein H2A.W blocks H1 deposition in WT. Loss of H2A.W then leads to overaccumulation of H1 at sites normally occupied by H2A.W, decreased accessibility, and reduced DNA methylation.

## Discussion

Chromosomal rearrangements are common in T-DNA insertion lines[34,35]. Here we identified a large chromosomal rearrangement in the *hta6-1* SALK line that resulted in a duplication of the translocated region during the generation of the *hta6-1 hta7 hta12* triple mutants (*h2a.w-1*). This chromosome rearrangement, and not the loss of H2A.W, is responsible for the developmental defects and CHG hypermethylation previously reported in *h2a.w-1*, as well as the lethality of *h2a.w-1 cmt3* quadruple mutants[22]. The absence of these defects in triple mutant *h2a.w-2* has now enabled us to analyze the direct impact of H2A.W on heterochromatin composition and accessibility.

H2A.W promotes higher order chromatin compaction[22] and increases nucleosome stability[26]. Hence, we expected loss of H2A.W to increase heterochromatin accessibility. Surprisingly, we observed the opposite, with accessibility decreasing in *h2a.w-2* in association with increased recruitment of H2A, H2A.X and H1 to

heterochromatin, indicating that H2A.W presence favors heterochromatin accessibility. H1 is known to stabilize the wrapping of DNA around the nucleosome, promote assembly of higher order chromatin structures[36], and influence nucleosome spacing[37,38]. In the presence of H2A.W, loss of H1 leads to increased accessibility and increased DNA methylation in heterochromatin. These phenotypes are enhanced in the absence of both H1 and H2A.W, suggesting that H2A.W also contributes to restricting heterochromatic DNA accessibility in the absence of H1. Gain of H1 in *h2a.w-2* heterochromatin correlates well with decreased accessibility in the same regions, suggesting that H2A.W also promotes accessibility in heterochromatin by restricting H1 levels. The antagonism between H1 and H2A.W may originate from a competition for linker DNA binding. The extended C-terminal tail of H2A.W interacts with linker DNA, and this interaction prevents micrococcal nuclease accessibility[26]. The H2A.W C-terminal tail is characterized by the SPKK motif[22], which binds A/T-rich DNA in its minor groove and causes condensation[39,40]. Among Arabidopsis H2A variants, the SPKK motif is unique to H2A.W[22]. Two SPKK-like motifs, SPAK and SP(G/A)K, are also present in the C-terminal tails of *Arabidopsis* H1.1 and H1.2 (Supplementary Fig. 9). The resulting competition between H2A.W and H1 for linker DNA binding could help prevent excessive H1 accumulation in heterochromatin. Because they do not contain SPKK motifs, H2A and H2A.X, which replace H2A.W in *h2a.w-2*, would not be able to compete with H1 for linker binding. Hence, H2A.W might allow some degree of nucleosome 'breathing' in compact heterochromatin, which would facilitate access of CMT3 and CMT2 to heterochromatic DNA, enabling the maintenance of DNA methylation patterns over these regions (Fig. 9). Although our data support the abovementioned scenario, we cannot exclude a potential impact of H2A and H2A.X enrichment in heterochromatin on DNA methylation and accessibility.

At genomic regions targeted by RdDM (DRM1/2-dependent regions and short TEs), which are located primarily outside of pericentromeric heterochromatin, non-CG methylation levels

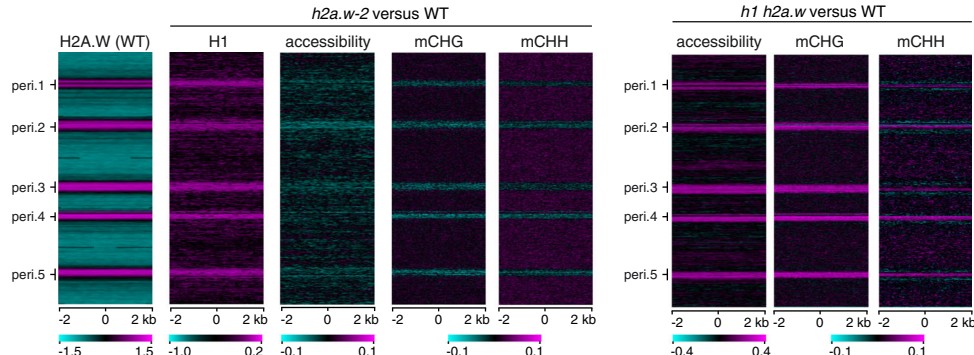

**Fig. 8 _h2a.w-2_ and _h1 h2a.w_ show opposite changes in accessibility and non-CG methylation.** The genome was divided into consecutive, non-overlapping 4-kb bins, which are stacked according to their genomic position from the top of chromosome 1 to the bottom of chromosome 5. Pericentromeric regions are indicated (peri.1 to peri.5). H2A.W enrichment in the WT (log2 ChIP/H3) is shown as a heat map on the left. Other heat maps show changes in H1 levels (_h2a.w-2_/WT; log2 ChIP/H3), changes in chromatin accessibility (log2 mutant/WT; ATAC-seq read depth), and changes in CHG and CHH DNA methylation (mutant minus WT) in _h2a.w-2_ versus WT and in _h1 h2a.w_ versus WT.

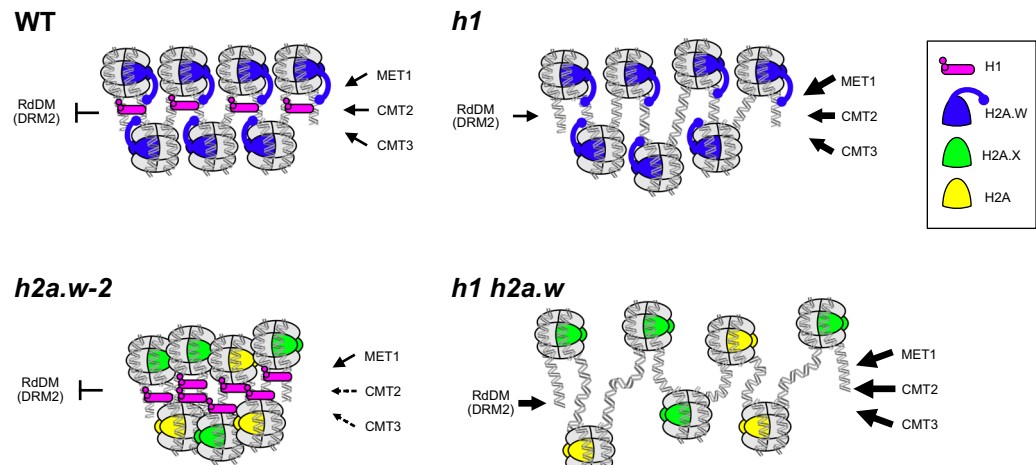

**Fig. 9 Model of co-regulation of heterochromatin accessibility by H2A.W and H1.** In WT, H2A.W and H1 both interact with heterochromatin linker DNA to maintain a normal balance of accessibility to chromatin modifiers and DNA methyltransferases in heterochromatin. In the absence of H2A.W, H1 over-accumulates at linker DNA and further reduces accessibility. Loss of H1 alone causes incomplete decompaction, while loss of both H1 and H2A.W causes strong decompaction and increased accessibility. Thus, H2A.W promotes chromatin compaction in the absence of H1, but also promotes heterochromatin 'breathing' by opposing excessive H1 incorporation. The thickness of the arrows illustrates the accessibility of MET1, CMT2, CMT3, and DRM2 to DNA. Dashed arrows represent the lowest accessibility, and for solid arrows, the thicker the arrow, the more accessible the heterochromatic DNA.

increase in _h2a.w-2_. These genomic regions are associated with H2A.W in WT. Because RdDM is inhibited by heterochromatin[3,30], this suggests that H2A.W provides these regions with an heterochromatic state/identity that is lost in _h2a.w-2_, thereby facilitating RdDM-mediated DNA methylation. In agreement with a previous report, we found that DRM1/2 target regions showed reduced levels of non-CG methylation in _h1_ (Fig. 7c)[3]. This loss of non-CG methylation was further aggravated in _h1 h2aw-2_ (Fig. 7c). It was proposed that in the absence of H1 these RdDM-dependent genomic regions become more accessible to enzymes that catalyze euchromatic histone modifications and antagonize DNA methylation[3]. We found that chromatin accessibility at these regions was similar in _h1_ and WT (Supplementary Fig. 10a). Furthermore, while DRM1/2-dependent regions are enriched in H2A.W, they are rather depleted in H1 (Supplementary Fig. 10b). Therefore, loss of H1 likely impacts DNA methylation at RdDM-dependent regions indirectly. We propose that increased heterochromatin accessibility in _h1_ allows RdDM to function efficiently in heterochromatin, thus depleting the RdDM machinery from its regular targets in chromosome arms. This interpretation is consistent with the recent report that

24-nucleotide sRNAs increase in heterochromatin but decrease at euchromatic TEs in _h1_ mutants[41]. As heterochromatin accessibility further increases when both H1 and H2A.W are lost, this results in a further loss of DNA methylation at usual RdDM targets in chromosome arms in _h1 h2aw-2_. Thus, maintenance of proper heterochromatin stability is also presumably important to restrain RdDM activity to specific genomic regions.

Although heterochromatin was long believed to be highly compact and inaccessible to transcriptional machinery, there is increasing evidence that low levels of accessibility within heterochromatin are required for proper heterochromatin formation by permitting access to various factors, including DNA and histone methyltransferases, that help maintain a heterochromatic state[42,43]. H2A.W is subject to specific modifications[44], and its dynamic deposition likely participates in the regulation of chromatin accessibility through its interaction with H1 and other yet unknown factors.

## Methods
**Plant material**. The _hta6_ (SALK_024544C), _hta7_ (GABI_149G05.01), _hta12_ (SAIL_667_D09), and _cmt3-11_ (SALK_148381) mutant lines used in this study

were all in the Col-0 genetic background. Plants were grown in long-day conditions (16 h light, 8 h dark) at 23 °C with 50% relative humidity.

**CRISPR-Cas9 targeted mutagenesis.** Design of optimal guide RNA (gRNA) sequences was performed using an online bioinformatic tool (https://www.genome.arizona.edu/crispr/index.html). The spacer (GTTTCGAAATCGATGAAAGC) was ligated between the two *BbsI* sites of the pEn-Chimera vector using annealed oligonucleotides (Supplementary Table 2) and then transferred by a single site Gateway LR reaction into the pDE-CAS9 binary vector. The detailed procedure and vectors are described in ref. [45]. Col-0 plants were transformed by floral dipping[46] and T1 transformants were isolated following BASTA selection. Identification of heritable targeted mutagenesis events was done by PCR amplification and sequencing of the region of interest. Two independent T2 lines were then selected that had segregated away the T-DNA coding for the gRNA and Cas9 expression cassette and contained a potential insertion of a single guanine 3 bp upstream of the protospacer adjacent motif (PAM) at the gRNA-targeted HTA6 5′ coding region. The +1 G insertion induces an early frame shift 89 bp downstream from the translation initiation site and a stop codon 195 bp downstream from the translation initiation site. Segregation of the mutant allele was analyzed in the T3 generation and we confirmed that both T2 lines were homozygous for the mutation. The knockout nature of this *hta6-2* allele was confirmed by immunoblot analysis using a specific antibody (see Fig. 1d). In subsequent crosses, a dCAPS assay was used to identify the *hta6-2* allele through a single *Bsa*BI digestion (cuts the mutant allele) of the PCR product (Supplementary Table 2).

**Southern blot.** Genomic DNA was extracted from rosette leaves using the Wizard® Genomic DNA Purification Kit (Promega) following manufacturer's instructions. 750 ng of DNA was digested overnight with 20 units of high fidelity *SspI* restriction enzyme (New England Biolabs) in Cutsmart® buffer and electrophoresed through a 1% agarose (w/v) gel for 8 h. The gel was depurinated (10 min in 0.25 N HCl), rinsed, denatured (30 min in 0.4 N NaOH, 1.5 M NaCl), neutralized (30 min in 0.5 M Tris-HCl, 1.5 M NaCl) and capillary blotted onto a Hybond-N + membrane (Amersham) overnight. The membrane was UV-crosslinked at 150 mJ. The DNA probe was amplified from Col-0 DNA with primers indicated in Supplementary Table 2, gel-purified, and labeled with α-$^{32}$P-dCTP using the random hexamer priming method (Megaprime DNA labeling system; Amersham) following manufacturer's instructions and subsequently purified on illustra MicroSpin S-200 HR columns (GE Healthcare Life Sciences). Hybridization was performed using the PerfectHyb™ Plus hybridization buffer (Sigma) following manufacturer's instructions, with overnight hybridization at 65 °C followed by one washing step (10 min) in 2X SSC 0.1% SDS and two washing steps (15 min each) in 0.5X SSC 0.1% SDS, all at 65 °C. The membrane was imaged on a Typhoon FLA 7000 (GE Healthcare Life Sciences).

**Inverse PCR.** A total of 250 ng of Col-0 and *hta6-1* genomic DNA were digested by *SspI* and then column-purified using the Neo Biotech gel extraction kit. To favor self-recircularization of the *SspI*-digested fragments, ligation was performed at 15 °C for 16 h using 100 ng of the digested DNAs and 4.5 U of T4 DNA ligase (Promega) in a final volume of 100 μl. Following column purification (Neo Biotech gel extraction kit), 1/50 of the eluted DNA was used as a template for a first round of inverse PCR (iPCR) with primers that closely match the expected extremity of the translocation. A second round of PCR amplification was done using nested primers and a 1/100th dilution of the first amplification as template. A specific product of around 2.1 kb was obtained for the *hta6-1* genomic DNA and sequenced (Eurofins) to identify the left border of the translocation. The primers used for iPCR are reported in Supplementary Table 2.

**Transcript analysis.** Total RNA was extracted with TRI Reagent (Sigma®) from 30 to 40 mg of fresh material following the manufacturer's instructions. 8 μg of RNA were treated for 1 h at 37 °C with 12 units of RQ1 DNase (Promega®) followed by phenol-chloroform extraction and ethanol precipitation of RNA which was subsequently dissolved in water. One-step reverse-transcription quantitative PCR (RT-qPCR) was performed with the SensiFAST™ SYBR® No-ROX One-Step kit (Bioline®) on an Eco™ Real-Time PCR System (Ilumina®) with the following program: 10 min at 45 °C, 5 min at 95 °C, and 40 cycles of 20 s at 95 °C and 30 s at 60 °C. A melting curve was generated at the end of the program to control for amplification specificity. Data was normalized to a reference gene and analyzed according to the $2^{-\Delta\Delta Ct}$ method. Means and standard errors of the mean were calculated from independent biological samples. Differences in the means for RT-qPCR data were tested using an unpaired Student's *t* test with Welch's correction with the *t.test* function of R version 3.4.0[47].

**Nuclear cytology.** Rosette leaves (fifth or sixth) from 3/4-week-old plants were fixed in 4% (v/v) formaldehyde/PBS 1X for 3 h at room temperature. Following fixation, leaves were placed between two layers of kitchen paper to remove excess buffer and a piece of approximately 0.5–1 cm$^2$ leaf tissue was chopped with a razor blade in 125 μL of extraction buffer from the CyStain UV Precise P kit (Partec) in a petri dish. Extracts were passed through a 30 μm filter to isolate nuclei and kept on ice. The procedure was repeated by adding 125 μL of extraction buffer to the petri

dish. After 2 min on ice, 20 μL of nucleus extract were supplemented with an equal volume of 60% acetic acid on a slide and stirred continuously with fine forceps on a 45 °C metal plate for 3 min; 60% acetic acid was added again and stirred for 3 min. The slide was cleared with an excess of an ethanol/acetic acid solution (3:1), air-dried and mounted with DAPI in Vectashield mounting medium (Vector Laboratories). Nuclei were visualized on a Zeiss Axio Imager Z1 epifluorescence microscope equipped with a PL Apochromat 100X/1.40 oil objective and images were captured with a Zeiss AxioCam MRm camera using the Zeiss ZEN software. The relative heterochromatin fraction was computed for each nucleus by calculating the ratio of the signal intensity at chromocenters over that of the entire nucleus using the ImageJ software (1.49 v).

**Nuclear protein extraction and immunoblot.** Nuclear protein extracts for Western blot analyses were prepared as described in ref. [23] with few modifications. For each sample 300 mg of 10-day old seedlings or 200 mg of floral buds (for H2A.W antibody characterization in Supplementary Fig. 1c) are frozen in liquid nitrogen and disrupted in 2 mL Eppendorf tubes using Qiagen TissueLyser II and metal beads to fine powder. Total ground powder is transferred into 15 mL falcon tube containing 5 mL of nuclei isolation buffer (NIB; 10 mM MES-KOH pH 5.3, 10 mM NaCl, 10 mM KCl, 250 mM sucrose, 2.5 mM EDTA, 2.5 mM ß-mercaptoethanol, 0.1 mM spermine, 0.1 mM spermidine, 0.3% Triton X-100) and supplemented with protease and phosphatase inhibitors (Roche), followed by vortexing until a fine suspension was obtained. The suspension was filtered through two layers of Miracloth into 50 mL Falcon tubes, followed by washing the Miracloth with 10 mL of NIB. Remaining buffer was carefully squeezed out of the Miracloth into the tube. Nuclei were pelleted by centrifugation at 1000 x *g* at 4 °C for 5 min. The pellet was washed once with 5 mL of NIB and centrifuged again. Nuclei were re-suspended in 1 mL of NIB and transferred to Eppendorf tubes followed by centrifugation for 5 min at 4 °C at maximum speed. Finally, nuclei were re-suspended in 150 μL of 1x PBS supplemented with protease and phosphatase inhibitors (Roche), mixed with 50 μL of 4x Laemmli loading buffer and boiled for 5 min. Once the samples reached room temperature, 2 μL of Benzonase (Millipore) was added and incubated for 10 min on bench. Samples are again boiled for 3 min to inactivate Benzonase. Samples were spun at maximum speed for 5 min to pellet down insoluble fraction and supernatant is transferred to fresh Eppendorf tubes. For Western blot analyses, 10 μL for histone variants and 5 μL for H3 (used as a loading control) were loaded per lane. Nuclear proteins were resolved using NuPAGE 4–12% Bis-Tris protein gels. Resolved proteins were transferred onto PVDF membrane using Bio-Rad wet transfer unit. blot analysis was performed using 1:1000 diluted antibodies in 5% milk in TBST. H2A.W.6, H2A, H2A.X and H2A.Z antibodies are affinity-purified rabbit polyclonal antibodies made by GenScript USA Inc (Piscataway, NJ) against peptides GGRKPPGAPKTKSVC, CPKKAGASKPSADE, CKVGKNKGDIGSASQ, and KPSGSDKDKDKKKPC, respectively[22]. H2A.W.7, and γH2A.X antibodies are reported in a previous study[23]. H2A.W and H1 antibodies were generated using peptides CTTKTPKSPSKATKSP and CRTGSSQYAIQKFIEEK, respectively, at Eurogentec.

**RNA-seq.** Total RNA was isolated with RNeasy Mini kit (Qiagen) from 10-day old seedlings in three replicates. DNase treatment was done on 2 μg of total RNA with DNA free DNase Kit (Invitrogen). From 1 μg of total RNA, rRNA was depleted using RiboZero kit (Illumina). NGS-libraries were generated using NEBnext Ultra II directional RNA library prep kit for Illumina and sequenced as PE75 reads on an Illumina NextSeq550.

Reads were trimmed and filtered for quality and adapter contamination using Trim Galore[48] and aligned to the TAIR10 genome using STAR[49]. Reads aligning equally well to more than one position in the reference genome were discarded, and probable PCR duplicates were removed using MarkDuplicates from the Picard Tools suite[50]. Alignment statistics for each library are available in Supplementary Table 3. Read counts for each gene and TE were obtained using htseq-count[51], with annotations from araport11[52]. Annotated TEs overlapping strongly (>80%) with an annotated TE gene were considered TE genes, and the TE annotation was discarded. Differential expression analysis was performed using DESeq2[53], and genes were considered differentially expressed with an adjusted *p*-value <0.05 and abs[log$_2$(fold change)] > 1.

**ATAC-seq.** ATAC-seq was performed as described in ref. [54]. Briefly, 0.5 g of freshly collected 10-day old seedlings was chopped in 4 mL of pre-chilled lysis buffer (15 mM Tris-HCl pH 7.5, 20 mM NaCl, 80 mM KCl, 0.5 mM spermine, 5 mM ß-mercaptoethanol, 0.2% Triton X-100). After chopping, the suspension was filtered through a 40 μM filter. Nuclei were further enriched using a sucrose gradient. Enriched nuclei were resuspended in 0.5 mL of pre-cooled lysis buffer with 4,6-Diamidino-2-Phenylindole (DAPI) and incubated for 15 min. DAPI-stained nuclei were sorted by FACS Aria III (BD Biosciences) with FACSDiva software (version 8). Sorted nuclei (50,000) were pelleted and washed once (10 mM Tris-HCl pH 8.0, 5 mM MgCl2). Tagmentation reaction was carried out using Nextera reagents (TDE1 Tagment DNA Enzyme (Catalog No. 15027865), TD Tagment DNA Buffer (Catalog No. 15027866)). Tagmented DNA was isolated using Qiagen MinElute PCR purification kit. NGS libraries were amplified using NEBNext high fidelity 2X master mix and Nextera primers. The number of PCR cycles was

determined using a method described in ref. [55]. NGS-libraries were PE75 sequenced on an Illumina NextSeq550.

Reads were trimmed and filtered as indicated above (RNA sequencing), and aligned to TAIR10 using bowtie2[56]. Reads aligning to multiple positions and PCR duplicates were removed (see RNA sequencing). Only properly paired reads were retained for the analysis. Alignment statistics for each library are available in Supplementary Table 4. Sample tracks and peaks in WT and *h2a.w-2* were obtained using Genrich[57] with parameters -p 0.01, -a 200, -l 100, and -g 100. ChrM, ChrC, and several rRNA regions with very high coverage were omitted from the analysis. Metaplots of ATAC-seq signal over various genomic regions were created using deeptools[58]. Plots of ATAC-seq signal over entire chromosomes are based on average signal over 1 kb non-overlapping bins tiled genome-wide, calculated using deeptools. Smoothed conditional mean of the signal was computed using the LOESS smoothing method with bin width span 0.1 and plotted using R[47]. A comparison of the ATAC-seq data generated in this study with selected published ATAC-seq datasets is provided in Supplementary Fig. 11.

**Whole genome bisulfite sequencing**. Genomic DNA was extracted from the aerial portions of 10-day old seedlings using the Wizard® Genomic DNA Purification Kit (Promega) following manufacturer's instructions. For WT and *h2a.w-2* replicates 1 and 2, sodium bisulfite conversion, library preparation, and sequencing on a Hiseq 4000 were performed at the Beijing Genomics Institute (Hong Kong) from 1 μg DNA, producing paired 100-bp (replicate 1) or 150-bp (replicate 2) paired-end reads. For remaining samples, methylated adapters were ligated prior to bisulfite treatment using the ZYMO EZ DNA methylation Gold kit, libraries were prepared using the Nugen ultralow methyl-seq kit and sequenced on NovaSeq 6000. Reads were trimmed, mapped to the TAIR10 genome, and methylation called using methylpy (version 1.4.3). Only uniquely mapping reads were retained. Alignment statistics for each library are available in Supplementary Table 5. Pericentromeres and chromosome arm regions were defined based on H3K9me2 distribution[59].

**ChIP-seq**. ChIP was performed as described in ref. [60]. Briefly, 3 g (approx. 0.3 mg for each immunoprecipitation (IP)) of 10-day old seedlings were fixed in 1% PFA. Fixed seedlings were ground to fine powder in liquid nitrogen using a mortar and pestle. Nuclei were isolated with M2 buffer (10 mM phosphate buffer pH 7.0, 100 mM NaCl, 10 mM ß-mercaptoethanol, 10 mM MgCl₂, 0.5% Triton X-100, 1 M hexylene glycol, 1× cOmplete protease inhibitor cocktail) and M3 buffer (10 mM phosphate buffer pH 7.0, 100 mM NaCl, 10 mM ß-mercaptoethanol, 1× cOmplete protease inhibitor cocktail). Chromatin shearing was done using a Covaris E220 with the following settings: treatment time 15 min, acoustic duty factor % 5.0, PIP 140 W, Cycles per burst 200 and max temperature 8 °C. IP, washes, and DNA isolation were carried out as described in ref. [60]. A total of 5 μl of each antibody (1 mg/mL) is used for IP. H2A, H2A.X and H2A.Z antibodies are reported in a previous study[22]. H3 (ab1791 Abcam), H3K9me1 (ab8896 Abcam), H3K9me2 (ab1220/Abcam), H3K27me1 (17-643/Millipore), and H1 (AS111801/Agrisera) antibodies were obtained from commercial sources. NGS libraries were generated using Ovation Ultralow Library System V2 (NuGEN) for replicate 1 and NEBNext Ultra II DNA preparation kit for replicate 2. NGS-libraries were SR75 sequenced on an Illumina NextSeq550.

Reads were trimmed, filtered, and aligned using bowtie2, and multi-mapping reads and PCR duplicates were removed, all as indicated above (see 'ATAC-seq' section). Alignment statistics for each library are available in Supplementary Table 6. Sample tracks and metaplots over genomic regions were obtained using deeptools[58] bamCoverage (–normalizeUsing CPM). All samples except for H3 were normalized to their matched H3 sample using deeptools bamCompare. Plots over entire chromosomes were obtained from average ChIP-seq signal over 1 kb non-overlapping bins tiled genome-wide and smoothed using the same approach as the ATAC-seq data. H2A.W ChIP-seq data were re-analyzed from ref. [22].

**Reporting summary**. Further information on research design is available in the Nature Research Reporting Summary linked to this article.

## Data availability

Data supporting the findings of this work are available within the paper and its Supplementary Information files. A reporting summary for this Article is available as a Supplementary Information file. The datasets and plant materials generated and analyzed during the current study are available from the corresponding author upon request. High throughput sequencing data has been deposited in the Gene Expression Omnibus (GEO) database and can be accessed with the accession number GSE146948. The source data underlying Figs. 1b, d, 4a, and 5a, as well as Supplementary Figs. 1, 5a, 6c and 7b are provided as a Source Data file.

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

## Acknowledgements

We thank James Watson from GMI for editing and comments on the manuscript. We thank the Vienna Biocenter Core Facility Next Generation Sequencing and we thank IMBA/IMP/GMI BioOptics facility for FACS sorting. Work in the Mathieu laboratory was supported by CNRS, Inserm, Université Clermont Auvergne, Young Researcher grants from the Auvergne Regional Council (to O.M.), an EMBO Young Investigator award (to O.M.), and a grant from the European Research Council (ERC, I2ST 260742 to O.M.). P.B. was supported by a PhD studentship from the Ministère de l'éducation nationale, de l'enseignement supérieur et de la recherche. Work in the Berger laboratory was supported by the Gregor Mendel Institute core funding from the Austrian Academy of Sciences and the Austrian Science Fund (FWF): I2303, P32054, P28320, and P26887. Work in the Jacobsen laboratory was supported by the National Institutes of Health (R35GM130272 to S.E.J.). C.L.P was supported by the National Institutes of Health under a Ruth L. Kirschstein National Research Service Award (F32GM136115). S.E.J. is an Investigator of the Howard Hughes Medical Institute. The funders had no role in study design, data collection and analysis, decision to publish, or preparation of the manuscript.

## Author contributions

P.B., C.L.P., R.Y., F.B, S.E.J., and O.M. designed the study. P.B., R.Y., T.P., Z.J.L., A.S, S.F., and M.N.P. performed experiments. C.L.P. and O.M performed bioinformatic analyses. P.B., C.L.P., R.Y., F.B., and O.M. wrote the manuscript. F.B., S.E.J., and O.M. coordinated the research.

## Competing interests

The authors declare no competing interests.
