## [Peer Review File · Nature Communications]

REVIEWER COMMENTS

Reviewer #1 (Remarks to the Author):

This is a study of the roles of histone variant H2A.W in heterochromatin regulation in Arabidopsis. The authors found out that there were some misinterpretations in the previous study (Yelagandula et al. 2014) due to unexpected chromosome rearrangement by T-DNA insertion in h2a.w-1. In this study, they report that H2A.W histones play roles in promoting chromatin accessibility while correcting some of the previous observations in Yelagandula et al. (2014). In the newly produced h2a.w-2 null mutant, H2A.W histones were depleted whereas the deposition of H2A.X and H2A was enhanced at pericentromeric regions. Contrary to expectations, h2a.w-2 showed decreased accessibility of chromatin and reduced DNA methylation at pericentromeric regions, which at least partly caused by increased H1 deposition. Overall data quality and organization look very good. Here are some points to improve the manuscript before publication.

- If there is no limitation to the length of introduction, it would be nice to introduce how each H2A, H2A.X and H2A.Z is deposited by chromatin remodelers or chaperones. In addition, more information about the roles of H2A and H2A.Z would be helpful, although H2A.X is well explained in the result part.
- Degrees of differential DNA methylation between WT and h2a.w-2 null mutant are relatively subtle. It seems H2A.W partly participates in DNA methylation pathways, limited to CHG and CHH contexts. Given the fact that h2a.w-2 is a null mutant, H2A.W seems to involve partially in DNA methylation pathways. The authors should clearly point out the partial loss of DNA methylation by h2a.w mutation.
- BS-seq data in Fig. 2a and 2b seem to present the average levels of two replicates. To see if each replicate is similar to the other, 1st and 2nd replicate data of h2a.w-2 could be shown as separate lines along with WT replicates in a supplementary figure.
- Authors mentioned H3 and H3K9me2 levels were unchanged in h2a.w-2. CHG DNA methylation is tightly associated with H3K9me2. What is explanation about this?
- The previous study in Yelagandula et al. 2014 showed H2A.W deposition was not affected in kyp suvh5 suvh6 and cmt3, mutants of H3K9 methyltransferases and CHG DNA methyltransferase, respectively. This could be mentioned to further address the relationship between H2A.W and H3K9me2 and CHG methylation.
- In Fig. 2b, CHH methylation is the most sensitive context by h2a.w-2 mutation. CHH levels were decreased at pericentromeric TEs whereas increased at euchromatic TEs where H1 is not usually enriched. Although it was argued H1 is the main reason for this, the authors need to mention if there are any potential effects of increased H2A and H2A.X levels at pericentromeric regions on DNA methylation.
- The discussion part emphasizes on H1 without explaining about misregulation of H2A.Z and H2A in h2a.w-2. Increases of H1 might be indirect. Loss of H2A.W might be more correlated with H2A.X and H2A than H1. Showing correlation values of each H1, H2A.X, and H2A to loss of H2A.W could be helpful to understand better.
- Is there any explanation why H2A.Z was unchanged in h2a.w-2?
- Another interesting finding is the opposing relationship between H2A.W and H1. In the discussion part, the authors mentioned "By competing with H1 for linker DNA binding and preventing excessive H1 accumulation, H2A.W might promote chromatin accessibility and nucleosome "breathing" in otherwise compact heterochromatin." As they mentioned, H2A.W and H1 include SPKK or SPKK-like motif. Are SPKK motifs found in H2A, H2A.X and H2A.Z?
- In page 8, "Profiles of epigenetic marks associated with heterochromatin, namely H3K9me1, H3K9me2, and H3K27me1, were similar in WT and h2a.w-2 (Supplementary Fig. 5a). Hence, maintenance of these post transcriptional modifications is independent of H2A.W..." Did they mean

"histone post translational modifications?"

- In page 10, "Interestingly, deposition of H1 in human osteoclasts depends on macroH2A, which is also localized specifically at heterochromatin and impacts its organization in mammals.." What's the relationship between H1 and macroH2A negative or positive?

Reviewer #2 (Remarks to the Author):

The question of how certain histone variants shape the chromatin landscape and instruct gene expression is an important topic in chromatin biology. The manuscript by Bourget et al. reports the role of the histone variant H2A.W in *Arabidopsis thaliana* which has previously been described in a report from Yelagandula et al., 2014 in *Cell*. However, the authors found that the observed phenotypes in the original h2a.w-1 triple mutant are due to a large genomic rearrangement in the hta6-1 T-DNA allele and not due to the loss of H2A.W function. It's really surprising and unfortunate that the original report in *Cell* has no complementation data which would have clearly indicated that there is an issue with the hta6-1 allele. The present manuscript is clearly written and I'm glad that the authors could partially correct their findings from the 2014 report. However, the newly generated h2a.w-2 triple mutant shows no developmental phenotype and only minor molecular phenotypes. In my opinion this manuscript should be considered for publication in *Nature Communications* after addressing some major concerns which are listed below:

Major concerns:

1. Functional complementation of the original h2a.w-1 triple mutant was missing in the *Cell* report and now a functional complementation of the new h2a.w-2 triple mutant is missing in this report. I'm not sure about the rationale of not presenting or not conducting complementation experiments, but in my opinion this manuscript should only be published when successful complementation can be shown. The successful complementation of the hta9hta11 with HTA11:HTA11-GFP (Kumar and Wigge, 2010 *Cell*) is a good example.
2. The quality of ChIP-seq datasets are hard to know just based on the plots provided. Genome browser shots should be presented at least for their H2A.Z ChIP-seq data. How do the H2A.Z ChIP-seq data and ATAC-seq data compare to publicly available datasets? Ideally, the authors should provide a genome browser link for the reviewers so that the quality of their data can be directly inspected.
3. The authors should better describe the antibodies used in their study. It's hard to judge their specificity from just a reference in Material & Methods section. For example, does the H2A antibody recognizes H2A.X?
4. The authors found an increase of H2A.X and replicative H2A in h2a.w-2 heterochromatic regions but did not discuss this further. Thus, the section seems relatively unconnected with the rest of the manuscript.
5. The antagonism between H2A.W-containing nucleosomes and H1 deposition is very interesting but wasn't fully mechanistically explored in this manuscript. h2a.w h1 pentuple mutants would have been helpful to analyze.
6. It is relatively difficult for readers that are not familiar with metaplots to really understand the author's findings. I think a final model that summarizes the findings/hypothesizes would be very helpful.

Minor points

1. There are too many subpanels in panel d of Fig.3 and the corresponding description is too unprecise. Pericentromeres and chromosome arms should be indicated more clearly.
2. Reference is missing for this sentence "Based on in vitro thermostability assays, replicative H2A confers higher stability than H2A.W and H2A.X, whereas H2A.Z nucleosomes are the least stable."
3. The last sentence of the abstract does not reflect the findings presented in the manuscript. It suggests that the authors discovered the mechanism by which H2A.W facilitates DNA methylation, fine tuning of H1 distribution, etc. This is not the case.

Reviewer #3 (Remarks to the Author):

This manuscript characterizes a newly-created h2a.w triple knockout in Arabidopsis. Importantly, it corrects the record from previous analysis which suffered from an uncharacterized genomic rearrangement. The analyses are logical and well-described, and the conclusions are supported by the data. The authors should be particularly commended for an extremely clear and well-written text.

The authors describe a genomic rearrangement/duplication linked to the previous T-DNA allele of hta6, and demonstrate that the rearrangement, not the loss of H2A.W, is responsible for the developmental phenotypes previously observed. The authors also create a new allele of hta6 using Cas9 genome editing. The evidence for the genomic rearrangement is strong and the authors perform careful genetic analysis to link this change to developmental phenotype.

The bulk of the manuscript is analysis of the newly-created h2a.w triple knockout mutant to understand the role of H2A.W. The authors assess DNA methylation, chromatin accessibility, and histone incorporation genome-wide. The mutant has decreased DNA methylation (particularly CHG and CHH methylation) in the pericentromere, but increased CHH methylation at DRM1/2-dependent sites in euchromatin. Although H2A.W is associated with inaccessible heterochromatin, the h2a.w knockout has reduced accessibility both in heterochromatin and sites of H2A.W incorporation in euchromatin, suggesting that its role is in fact to increase accessibility of heterochromatin by inhibiting the association of H1.

The manuscript is in good shape, and I have only a couple of suggestions for improvement.

1. A critical aspect of many of the analyses is the breakdown between "pericentromeric" and "euchromatic" regions. A description of how these were delineated should be included in the Methods.
2. In Fig 5e the scale for the methylation plots (from -0.01 to 0) are small in comparison to the kernel density plot in Figure 2e, which shows that this value should vary from -0.2 to 0.4. What explains this discrepancy? In particular, the lack of any signal above 0 suggests that there are no windows with increased CHG or CHH methylation in h2a.w-2, counter to conclusions earlier in the manuscript.

POINT-BY-POINT RESPONSE TO REVIEWER'S COMMENTS

Reviewer #1 (Remarks to the Author):

This is a study of the roles of histone variant H2A.W in heterochromatin regulation in Arabidopsis. The authors found out that there were some misinterpretations in the previous study (Yelagandula et al. 2014) due to unexpected chromosome rearrangement by T-DNA insertion in h2a.w-1. In this study, they report that H2A.W histones play roles in promoting chromatin accessibility while correcting some of the previous observations in Yelagandula et al. (2014). In the newly produced h2a.w-2 null mutant, H2A.W histones were depleted whereas the deposition of H2A.X and H2A was enhanced at pericentromeric regions. Contrary to expectations, h2a.w-2 showed decreased accessibility of chromatin and reduced DNA methylation at pericentromeric regions, which at least partly caused by increased H1 deposition. Overall data quality and organization look very good. Here are some points to improve the manuscript before publication.

Preliminary statement:

Please note that in response to reviewers' 2 point #5, we generated h1 h2a.w-2 quintuple mutants and determined their profiles of DNA methylation and chromatin accessibility. In the course of these analyses, we found that the decrease in accessibility of h2a.w-2 was not as pronounced in these new libraries. Compared with previously published ATAC-seq data, we found that the ATAC-seq libraries used in the original submission had a lower signal-to-noise ratio and we therefore decided to exclude them from this new revised version.

We now report a dramatic increase in the accessibility and DNA methylation in heterochromatin of h2a.w-2 h1 quintuple mutants relative to h1 mutants (these new data are presented in figs 6, 7, 8 and described in a new paragraph at the end of the "results" section). These new findings further support our original interpretation that increased histone H1 incorporation is responsible for decreased accessibility and DNA methylation in h2a.w-2 mutants. Additionally, these new data highlight that, although H2A.W and H1 compete for heterochromatin occupancy, they co-regulate heterochromatin accessibility and DNA methylation.

- If there is no limitation to the length of introduction, it would be nice to introduce how each H2A, H2A.X and H2A.Z is deposited by chromatin remodelers or chaperones. In addition, more information about the roles of H2A and H2A.Z would be helpful, although H2A.X is well explained in the result part.

As suggested, we have added more details about the H2A variants to the introduction, mentioning their roles, genomic distribution and chaperones where relevant.

- Degrees of differential DNA methylation between WT and h2a.w-2 null mutant are relatively subtle. It seems H2A.W partly participates in DNA methylation pathways, limited to CHG and CHH contexts. Given the fact that h2a.w-2 is a null mutant, H2A.W seems to involve partially in DNA methylation pathways. The authors should clearly point out the partial loss of DNA methylation by h2a.w mutation.

We agree and this was actually the message we intended to convey. We have modified the manuscript to make this conclusion clearer.

- BS-seq data in Fig. 2a and 2b seem to present the average levels of two replicates. To see if each replicate is similar to the other, 1st and 2nd replicate data of h2a.w-2 could be shown as separate lines along with WT replicates in a supplementary figure.

The two BS-seq replicates of h2a.w-2 and respective wild types are highly similar. As requested, we now show these separately in supplementary figure 3.

- Authors mentioned H3 and H3K9me2 levels were unchanged in h2a.w-2. CHG DNA methylation is tightly associated with H3K9me2. What is explanation about this?

This is indeed an intriguing observation. As now mentioned in the revised manuscript (page 8, lines 271-275), we think this suggests that SUVH4/5/6 may be less sensitive than CMT2/3 to changes in accessibility occurring in h2a.w-2 mutants, and/or that decrease in H3K9me2 is below the detection threshold allowed by ChIP-seq.

- The previous study in Yelagandula et al. 2014 showed H2A.W deposition was not affected in kyp suvh5 suvh6 and cmt3, mutants of H3K9 methyltransferases and CHG DNA methyltransferase, respectively. This could be mentioned to further address the relationship between H2A.W and H3K9me2 and CHG methylation.

We appreciate the reviewer's suggestion and now mention that H2A.W incorporation does not require CHG methylation in the results section (page 6, line 196).

- In Fig. 2b, CHH methylation is the most sensitive context by h2a.w-2 mutation. CHH levels were decreased at

pericentromeric TEs whereas increased at euchromatic TEs where H1 is not usually enriched. Although it was argued H1 is the main reason for this, the authors need to mention if there are any potential effects of increased H2A and H2A.X levels at pericentromeric regions on DNA methylation.

H1 is thought to inhibit DNA methyltransferases access to heterochromatic DNA, therefore we propose that the moderate decrease in non-CG methylation at heterochromatic TEs is a consequence of increased H1 levels in h2a.w-2 heterochromatin. Nonetheless, we agree with the reviewer that we cannot exclude potential effects of increased H2A.X and H2A levels at pericentromeric heterochromatin on DNA methylation. As requested, this is now clearly mentioned in the revised manuscript (see page 11, lines 368-370).

RdDM is known to be inhibited by heterochromatin and RdDM preferentially acts on short, less heterochromatic, TEs. We now show that RdDM-dependent regions are rather depleted in H1 but enriched in H2A.W (see supplementary fig 10). Because H2A.W is a heterochromatin mark, we propose that increased CHH methylation at these regions in h2a.w is a direct consequence of H2A.W loss. We propose that loss of CHH methylation at RdDM regions in h1 is an indirect consequence of the retargeting of RdDM to pericentromeric heterochromatin, which, as we now show, becomes more accessible when H1 is lost (see supplementary fig 10). Consistent with this hypothesis, our new data show that combined loss of H1 and H2A.W drastically enhances pericentromeric heterochromatin accessibility and is associated with a further loss of non-CG methylation at RdDM targets (see fig 6, 7, 8).

- The discussion part emphasizes on H1 without explaining about misregulation of H2A.Z and H2A in h2a.w-2. Increases of H1 might be indirect. Loss of H2A.W might be more correlated with H2A.X and H2A than H1. Showing correlation values of each H1, H2A.X, and H2A to loss of H2A.W could be helpful to understand better.

Our data are consistent with a model wherein changes in H1 levels are responsible for the changes in heterochromatin accessibility and associated changes in DNA methylation in h2a.w-2 mutants. Because H2A.W variants and linker histone H1 both contain SPKK-like motifs and are able to bind linker DNA, we believe that lack of H2A.W directly allows for more H1 binding into h2a.w-2 heterochromatin.

Other H2A.W variants must compensate for the absence of H2A.W in heterochromatin to maintain the integrity of nucleosomes. Our data shows that H2A and H2A.X replace H2A.W in pericentromeres in h2a.w-2 mutants, but

H2A.Z is not affected. As requested by the reviewer, we computed correlation coefficients of changes in H1, H2A.X, and H2A in h2a.w-2 to WT H2A.W levels (loss of H2A.W is complete in h2a.w-2, so that correlation to loss of H2A.W cannot be computed). Changes in H2A and H2A.X appear much more correlated with WT H2A.W than changes in H1 levels do (see plots on the left, R indicate Pearson correlation coefficients). However, this is expected given that H2A and H2A.X are nearly absent from WT heterochromatin, while H1 is already enriched in these regions in the WT. Therefore, we believe these correlation values are not really informative and would prefer not to include these in the manuscript.

We believe that our new data on h1 and h1h2a.w mutants further support our initial interpretation on the importance of H1 changes. Nonetheless, we fully acknowledge that it would be of interest to assess if replacement of H2A.W by H2A and H2A.X in h2a.w heterochromatic nucleosomes affects heterochromatin maintenance. We have begun generating h2a.w h2a.x quintuple mutants to test this; however, these efforts have been significantly delayed by the fact that we again identified a genomic rearrangement in the published h2a.x mutant line, leading us to generate new h2a.x mutants using CRISPR/Cas9. Due to the necessity of breeding for new mutant combinations, such analysis would significantly delay the current submission.

- Is there any explanation why H2A.Z was unchanged in h2a.w-2?

There is global anticorrelation between CG DNA methylation and H2A.Z, and it was shown to be primarily caused by the exclusion of H2A.Z from methylated DNA (Coleman-Derr and Zilberman 2012; Zilberman et al., 2008). That H2A.Z remains largely unchanged in h2a.w-2 is in good agreement with the fact that CG methylation patterns are also largely unaltered in the mutant. This is now mentioned in the revised manuscript.

- Another interesting finding is the opposing relationship between H2A.W and H1. In the discussion part, the authors mentioned "By competing with H1 for linker DNA binding and preventing excessive H1 accumulation, H2A.W might promote chromatin accessibility and nucleosome "breathing" in otherwise compact heterochromatin." As they mentioned, H2A.W and H1 include SPKK or SPKK-like motif. Are SPKK motifs found in H2A, H2A.X and H2A.Z?

The SPKK motif is specific to the H2A.W variants. We added a sentence to explicitly mention this.

- In page 8, "Profiles of epigenetic marks associated with heterochromatin, namely H3K9me1, H3K9me2, and H3K27me1, were similar in WT and h2a.w-2 (Supplementary Fig. 5a). Hence, maintenance of these post transcriptional modifications is independent of H2A.W..." Did they mean "histone post translational modifications?"

We corrected the sentence.

- In page 10, "Interestingly, deposition of H1 in human osteoclasts depends on macroH2A, which is also localized specifically at heterochromatin and impacts its organization in mammals.." What's the relationship between H1 and macroH2A negative or positive?

Kim et al. (Oncogene 2018) proposed a model where macroH2A recruits H1.2 to chromatin. However, given the new data provided in the revised manuscript, we revised our model and removed this part from the discussion.

Reviewer #2 (Remarks to the Author):

The question of how certain histone variants shape the chromatin landscape and instruct gene expression is an important topic in chromatin biology. The manuscript by Bourget et al. reports the role of the histone variant H2A.W in *Arabidopsis thaliana* which has previously been described in a report from Yelagandula et al., 2014 in Cell. However, the authors found that the observed phenotypes in the original h2a.w-1 triple mutant are due to a large genomic rearrangement in the hta6-1 T-DNA allele and not due to the loss of H2A.W function. It's really surprising and unfortunate that the original report in Cell has no complementation data which would have clearly indicated that there is an issue with the hta6-1 allele. The present manuscript is clearly written and I'm glad that the authors could partially correct their findings from the 2014 report. However, the newly generated h2a.w-2 triple mutant shows no developmental phenotype and only minor molecular phenotypes. In my opinion this manuscript should be considered for publication in Nature Communications after addressing some major concerns which are listed below:

Major concerns:

1. Functional complementation of the original h2a.w-1 triple mutant was missing in the Cell report and now a functional complementation of the new h2a.w-2 triple mutant is missing in this report. I'm not sure about the rationale of not presenting or not conducting complementation experiments, but in my opinion this manuscript should only be published when successful complementation can be shown. The successful complementation of the hta9hta11 with HTA11:HTA11-GFP (Kumar and Wigge, 2010 Cell) is a good example.

We introduced a HTA6 genomic construct under control of the endogenous HTA6 promoter in the h2a.w-2 triple mutant and used these lines to quantify chromocenter decompaction and transcript accumulation at selected genes upregulated in h2a.w-2 mutants. We found that the HTA6 genomic construct partially rescued the defects in heterochromatin condensation and gene expression provoked by the h2a.w-2 mutations. These new data are shown in the new supplementary Fig. 5 and mentioned in the main text.

2. The quality of ChIP-seq datasets are hard to know just based on the plots provided. Genome browser shots should be presented at least for their H2A.Z ChIP-seq data. How do the H2A.Z ChIP-seq data and ATAC-seq data compare to publicly available datasets? Ideally, the authors should provide a genome browser link for the reviewers so that the quality of their data can be directly inspected.

Our data can be accessed from the GEO repository using the record number GSE146948 with the following secure token: gjqbsoyoxwlllex

Bigwig tracks are provided for easy visualization in genome browsers.

Following reviewer's request, we included genome browser screenshots in supplementary Fig. 11, showing that our H2A.Z datasets compare favorably with previously published data. We also include spearman correlation coefficients of the comparison of our WT ATAC-seq samples with several available WT datasets, showing that our data are highly similar to published data.

3. The authors should better describe the antibodies used in their study. It's hard to judge their specificity from just a reference in Material & Methods section. For example, does the H2A antibody recognizes H2A.X?

We supplemented the Methods section with information about the peptides used to generate the antibodies. Our previous characterization of the H2A antibody (Yelagandula et al. 2014 Cell) showed that H2A does not recognize H2A.X.

4. The authors found an increase of H2A.X and replicative H2A in h2a.w-2 heterochromatic regions but did not discuss this further. Thus, the section seems relatively unconnected with the rest of the manuscript.

Our reasoning for conducting these experiments was to determine which H2A variant(s) replaced H2A.W in h2a.w-2 heterochromatin. This was of importance as H2A variants confer distinct stability to nucleosomes, and different H2A variants are more or less favorable to DNA methylation. Particularly, H2A.Z nucleosomes are the least stable, and H2A.Z incorporation is antagonistic with DNA methylation (Coleman-Derr and Zilberman, 2012; Zilberman et al, 2008), so that replacement of H2A.W by H2A.Z may have been responsible for the DNA methylation and accessibility changes observed in h2a.w. We rephrased this section of the manuscript to clarify our reasoning for conducting these experiments.

We fully acknowledge that it would be of interest to assess if replacement of H2A.W by H2A and H2A.X in h2a.w heterochromatic nucleosomes affects heterochromatin maintenance. We have begun generating h2a.w h2a.x quintuple mutants to test this; however, these efforts have been significantly delayed by the fact that we again identified a genomic rearrangement in the published h2a.x mutant line, leading us to generate new h2a.x mutants using CRISPR/Cas9. Due to the necessity of breeding for new mutant combinations, such analysis would significantly delay the current submission.

5. The antagonism between H2A.W-containing nucleosomes and H1 deposition is very interesting but wasn't fully mechanistically explored in this manuscript. h2a.w h1 pentuple mutants would have been helpful to analyze.

We thank the reviewer for this suggestion. In this revised manuscript, we have further explored the relationship between H2A.W and H1. We generated h2a.w-2 h1 quintuple mutants and determined profiles of DNA methylation and chromatin accessibility to DNA (including h1, h2a.w-2 and WT as controls). In the course of these analyses, we found that the decrease in accessibility of h2a.w-2 was not as pronounced in these new libraries. Compared with previously published ATAC-seq data, we found that the ATAC-seq libraries used in the original submission had a lower signal-to-noise ratio and we therefore decided not to include these in the revised version.

We now report a dramatic increase in the accessibility and DNA methylation in heterochromatin of h2a.w-2 h1 quintuple mutants relative to h1 mutants (these new data are presented in figs 6, 7, 8 and described in a new paragraph at the end of the "results" section). These new findings further support our original interpretation that increased histone H1 incorporation is responsible for decreased accessibility and DNA methylation in h2a.w-2 mutants. Additionally, these new data highlight that, although H2A.W and H1 compete for heterochromatin occupancy, they co-regulate heterochromatin accessibility and DNA methylation.

6. It is relatively difficult for readers that are not familiar with metaplots to really understand the author's findings. I think a final model that summarizes the findings/hypothesizes would be very helpful.

A model summarizing our findings and hypotheses is provided as a new figure 9.

Minor points

1. There are too many subpanels in panel d of Fig.3 and the corresponding description is too unprecise. Pericentromeres and chromosome arms should be indicated more clearly.

Figure 3 has been simplified and no longer contains subpanels. The pericentromeric region is now represented as a grey rectangle in the whole chromosome plot in Fig 3b.

2. Reference is missing for this sentence "Based on in vitro thermostability assays, replicative H2A confers higher stability than H2A.W and H2A.X, whereas H2A.Z nucleosomes are the least stable."

Thank you. This mistake has been corrected.

3. The last sentence of the abstract does not reflect the findings presented in the manuscript. It suggests that the authors discovered the mechanism by which H2A.W facilitates DNA methylation, fine tuning of H1 distribution, etc. This is not the case.

We rephrased the abstract to describe our findings more precisely.

Reviewer #3 (Remarks to the Author):

This manuscript characterizes a newly-created h2a.w triple knockout in Arabidopsis. Importantly, it corrects the record from previous analysis which suffered from an uncharacterized genomic rearrangement. The analyses are logical and well-described, and the conclusions are supported by the data. The authors should be particularly commended for an extremely clear and well-written text.

The authors describe a genomic rearrangement/duplication linked to the previous T-DNA allele of hta6, and demonstrate that the rearrangement, not the loss of H2A.W, is responsible for the developmental phenotypes previously observed. The authors also create a new allele of hta6 using Cas9 genome editing. The evidence for the genomic rearrangement is strong and the authors perform careful genetic analysis to link this change to developmental phenotype.

The bulk of the manuscript is analysis of the newly-created h2a.w triple knockout mutant to understand the role of H2A.W. The authors assess DNA methylation, chromatin accessibility, and histone incorporation genome-wide. The mutant has decreased DNA methylation (particularly CHG and CHH methylation) in the pericentromere, but increased CHH methylation at DRM1/2-dependent sites in euchromatin. Although H2A.W is associated with inaccessible heterochromatin, the h2a.w knockout has reduced accessibility both in heterochromatin and sites of H2A.W incorporation in euchromatin, suggesting that its role is in fact to increase accessibility of heterochromatin by inhibiting the association of H1.

The manuscript is in good shape, and I have only a couple of suggestions for improvement.

We are grateful to the reviewer for this very positive assessment of our work.

Preliminary statement:

Please note that in response to reviewers' 2 point #5, we generated h1 h2a.w-2 quintuple mutants and determined their profiles of DNA methylation and chromatin accessibility. In the course of these analyses, we found that the decrease in accessibility of h2a.w-2 was not as pronounced in these new libraries. Compared to previously published ATAC-seq data, we found that the ATAC-seq libraries used in the original submission had a lower signal-to-noise ratio and we therefore decided to exclude them from this new revised version.

We now report a dramatic increase in accessibility and DNA methylation in heterochromatin of h2a.w-2 h1 quintuple mutants relative to h1 mutants (these new data are presented in figs 6, 7, 8 and described in a new paragraph at the end of the "results" section). These new findings further support our original interpretation that increased histone H1 incorporation is responsible for decreased accessibility and DNA methylation in h2a.w-2 mutants. Additionally, these new data highlight that, although H2A.W and H1 compete for heterochromatin occupancy, they co-regulate heterochromatin accessibility and DNA methylation.

1. A critical aspect of many of the analyses is the breakdown between "pericentromeric" and "euchromatic" regions. A description of how these were delineated should be included in the Methods.

Pericentromeric and euchromatic chromosomal regions were defined based on their association level with H3K9me2, as first done by Bernatavichute et al. (2008). This is now mentioned in the Methods section of the revised manuscript (page 15, line 509).

2. In Fig 5e the scale for the methylation plots (from -0.01 to 0) are small in comparison to the kernel density plot in Figure 2e, which shows that this value should vary from -0.2 to 0.4. What explains this discrepancy? In particular, the lack of any signal above 0 suggests that there are no windows with increased CHG or CHH methylation in h2a.w-2, counter to conclusions earlier in the manuscript.

The heatmaps of DNA methylation differences in the original Fig 5e were generated using erroneous parameters, we thank the reviewer for drawing our attention to this problem. Corrected heatmaps are now presented in the new revised figure 8.

Please note that kernel density plots in Fig 2d and Fig 7b,c represent density of methylation changes over a limited number of genomic regions (13,167 for CMT2-dependent regions and 6,851 for DRM2-dependent regions), whereas Fig 8 shows average methylation changes along the whole genome. This explains the different scales.

REVIEWERS' COMMENTS

Reviewer #1 (Remarks to the Author):

The authors made appropriate and careful additional experiments and revisions. In the end, H1 seems to be more major component for chromatin compaction. This is supported by the new datasets showing the gains of DNA methylation and chromatin accessibility in h2a.w h1 mutant as opposed to the results of h2a.w. What is clear here is that H2A.W can be a back-up system of heterochromatin maintenance when H1 protein is absent. Unfortunately, the roles of H2A.W in heterochromatin regulation are not still clear. However, this would be a complementary and important report in the area of plant chromatin biology.

- It seems that h2a.w h1 might have some phenotypic impacts since chromatin accessibility and DNA methylation were changed at some levels. Addition of the pictures of h2a.w h1 along with appropriate control genotypes is recommended.

- Figure 9: Model figure legend does not explain about arrows. For example, what are the dashed arrows in h2a.w?

Reviewer #2 (Remarks to the Author):

The authors have appropriately addressed my concerns. Although it only shows a partial rescue the new complementation data is acceptable. In short, the manuscript is acceptable for publication Nature Communications.

Reviewer #3 (Remarks to the Author):

This is an excellent manuscript and deserving of immediate publication. The work corrects the public record regarding phenotypes associated with histone variant H2A.W and moves our understanding of chromatin dynamics forward significantly. The manuscript includes a vast array of approaches to understand histone variant H2A.W, including ChIP-seq, RNA-seq, ATAC-seq, and BS-seq in a variety of mutant backgrounds. The addition of the quintuple h1 h2a.w mutant as suggested by reviewer 2 is a very nice addition. The analysis of this huge amount of data is carefully done and the figures are very clear. I particularly appreciate the distinction between heterochromatin and euchromatin that is carried throughout the paper, and discussion of how changes in one domain can have indirect consequences on the other domain through retargeting of methylation machinery, for example. I have no further suggestions for improvement.

POINT-BY-POINT RESPONSE TO REVIEWER'S COMMENTS

Reviewer #1 (Remarks to the Author):

The authors made appropriate and careful additional experiments and revisions. In the end, H1 seems to be more major component for chromatin compaction. This is supported by the new datasets showing the gains of DNA methylation and chromatin accessibility in h2a.w h1 mutant as opposed to the results of h2a.w. What is clear here is that H2A.W can be a back-up system of heterochromatin maintenance when H1 protein is absent. Unfortunately, the roles of H2A.W in heterochromatin regulation are not still clear. However, this would be a complementary and important report in the area of plant chromatin biology.

- It seems that h2a.w h1 might have some phenotypic impacts since chromatin accessibility and DNA methylation were changed at some levels. Addition of the pictures of h2a.w h1 along with appropriate control genotypes is recommended.

We have added representative pictures of h2a.w h1, h2a.w, h1 and wild-type plants to Fig. 6a.

- Figure 9: Model figure legend does not explain about arrows. For example, what are the dashed arrows in h2a.w?

A description of the arrows is now provided in the legend of Fig. 9.

Reviewer #2 (Remarks to the Author):

The authors have appropriately addressed my concerns. Although it only shows a partial rescue the new complementation data is acceptable. In short, the manuscript is acceptable for publication Nature Communications.

Thank you

Reviewer #3 (Remarks to the Author):

This is an excellent manuscript and deserving of immediate publication. The work corrects the public record regarding phenotypes associated with histone variant H2A.W and moves our understanding of chromatin dynamics forward significantly. The manuscript includes a vast array of approaches to understand histone variant H2A.W, including ChIP-seq, RNA-seq, ATAC-seq, and BS-seq in a variety of mutant backgrounds. The addition of the quintuple h1 h2a.w mutant as suggested by reviewer 2 is a very nice addition. The analysis of this huge amount of data is carefully done and the figures are very clear. I particularly appreciate the distinction between heterochromatin and euchromatin that is carried throughout the paper, and discussion of how changes in one domain can have indirect consequences on the other domain through retargeting of methylation machinery, for example. I have no further suggestions for improvement.

Thank you